# Review of Integrated Chassis Control Techniques for Automated Ground Vehicles

**DOI:** 10.3390/s24020600

**Published:** 2024-01-17

**Authors:** Viktor Skrickij, Paulius Kojis, Eldar Šabanovič, Barys Shyrokau, Valentin Ivanov

**Affiliations:** 1Transport and Logistics Competence Centre, Transport Engineering Faculty, Vilnius Gediminas Technical University, 10223 Vilnius, Lithuania; 2Department of Mobile Machinery and Railway Transport, Transport Engineering Faculty, Vilnius Gediminas Technical University, 10105 Vilnius, Lithuania; 3Department of Cognitive Robotics, Delft University of Technology, 2628 CD Delft, The Netherlands; 4Smart Vehicle Systems—Working Group, Technische Universität Ilmenau, Ehrenbergstr, 15, 98693 Ilmenau, Germany

**Keywords:** automated driving, electric vehicles, integrated chassis control, vehicle dynamics, vehicle state estimation, control allocation, sensors

## Abstract

Integrated chassis control systems represent a significant advancement in the dynamics of ground vehicles, aimed at enhancing overall performance, comfort, handling, and stability. As vehicles transition from internal combustion to electric platforms, integrated chassis control systems have evolved to meet the demands of electrification and automation. This paper analyses the overall control structure of automated vehicles with integrated chassis control systems. Integration of longitudinal, lateral, and vertical systems presents complexities due to the overlapping control regions of various subsystems. The presented methodology includes a comprehensive examination of state-of-the-art technologies, focusing on algorithms to manage control actions and prevent interference between subsystems. The results underscore the importance of control allocation to exploit the additional degrees of freedom offered by over-actuated systems. This paper systematically overviews the various control methods applied in integrated chassis control and path tracking. This includes a detailed examination of perception and decision-making, parameter estimation techniques, reference generation strategies, and the hierarchy of controllers, encompassing high-level, middle-level, and low-level control components. By offering this systematic overview, this paper aims to facilitate a deeper understanding of the diverse control methods employed in automated driving with integrated chassis control, providing insights into their applications, strengths, and limitations.

## 1. Introduction

One of the distinctive features of modern vehicles on the architectural level is an increased number of active and X-by-wire chassis components involved in many safety- and comfort-related tasks. At the same time, various chassis systems can perform quite similar functions, e.g., the vehicle trajectory can be corrected by the electronic stability control (ESC) with brake actuators, torque vectoring (TV) with multiple electric motors, or active front/rear steering. This suggests that it is possible to integrate several subsystems to implement a coordinated control in one or more vehicle performance qualities, which leads to the concept of integrated chassis control (ICC). Due to various relevant terms that can be found in the literature to describe this concept, within the framework of this paper, the following definition can be proposed: *An integrated chassis control is a complex system composed of several active sub-systems like brakes, steering, suspension, individual-wheel electric motors, etc., which can be controlled in a coordinated way to enhance vehicle dynamics in general with a simultaneous improvement of various vehicle characteristics like driving safety, comfort, and energy efficiency*. 

The proposed definition is also similar to interpretations from other studies, where the ICC is associated with a system designed to coordinate the control actions of individual chassis (and optionally powertrain) actuators, to achieve superior vehicle performance according to various criteria [1,2].

The sub-systems that can be included in the ICC are brakes, electric motors, active suspension variants [2,3], and active steering [4,5]. In some studies, less widespread components such as active aerodynamics [2], wheel positioning systems [6,7], anti-roll bars [8], and dynamic tyre pressure control [9] are also considered within the ICC context. Hence, diverse actuator combinations are possible, making the ICC an over-actuated system. In this case, an improper integration of several sub-systems leads to overlapping regions of control. Therefore, to handle such an over-actuation and prevent control objective interference between sub-systems, an integration algorithm is required to optimally allocate the control actions between the involved actuators.

Although ICC concepts first appeared in studies published in the late 1980s [10,11], for a long time, the implementation of this approach has been limited to vehicles of the premium and high-performance classes. However, the topic of ICC has received increased research attention over the past decade due to the electrification and automation of road vehicles. First, mature and cost-efficient technologies for highly dynamic actuators such as X-by-wire systems and on-board/in-wheel electric motors (IWM) can be identified as a deciding factor for the availability of ICC for a broad spectrum of vehicle classes. Second, ICC can contribute to the solutions to the following essential challenges:for electric vehicles (EV):
energy-efficient drivingimproved driveabilitybetter ride quality, particularly for EVs with IWM
for automated driving (AD):
redundancynear-to-ideal driving comfortprecise motion control


Automated and electric ground vehicles demonstrate the demand for ICC from different points of view.

The main contributions of this paper are:Systematising knowledge of ICC targeting path tracking (PT) as part of AD: This paper provides a comprehensive synthesis of existing knowledge related to ICC, specifically tailored to the critical task of PT in the context of AD.Presenting a systematic overview of applied control methods: Building on the systematisation of knowledge, this paper delivers a structured and systematic overview of the various control methods used in ICC and PT. This includes a detailed examination of sensing, state estimation techniques, reference generation strategies, and the hierarchy of controllers, encompassing high-level, middle-level, and low-level control components. By offering this systematic overview, this paper facilitates a deeper understanding of the diverse control methods employed in ICC, providing insights into their applications, strengths, and limitations.

The rest of this paper consists of four sections. In Section 2, a literature review is performed. In Section 3, a typical control layout and block structure and functionality are presented. In Section 4, applied control methods are presented including parameter estimation, reference generation, and high-level and middle-level controllers. In Section 5, a discussion and conclusions are presented.

## 2. Previous Studies

Based on extensive research over the past decades, ICC systems can be divided into five categories:(i).systems with longitudinal dynamics integration(ii).systems with combined longitudinal and lateral integration(iii).systems with lateral and vertical components(iv).systems with longitudinal and vertical integration(v).systems with longitudinal, lateral, and vertical integration

The first category involves systems integrating longitudinal vehicle dynamics. One example is the brake blending in EV [12,13]. In this scenario, friction brakes work together with electric motors in a regeneration mode to ensure the required braking performance of the vehicle while improving energy efficiency. Another example of ICC results from the coordinated operation of ESC with friction brakes and TV. This coordinated use positively affects the vehicle’s agility and increases the tyre friction utilisation.

The second category involves systems with longitudinal and lateral integration. For example, the fusion of active steering and braking systems has been proposed in various studies [5,14,15,16,17,18,19,20,21,22,23,24,25,26,27,28]. In the past decade, due to more attention to EV dynamics control, the combination of active steering with the propulsion system has also attracted attention [29,30]. In such an integration, the steering is the primary tool for lateral dynamics control, while braking serves as a complementary measure for vehicle stabilisation [31]. Further studies, [32,33], utilised active rear-axis steering and brakes to enhance a vehicle’s yaw stability. In several relevant studies, the active control of four-wheel steering combined with brakes has been chosen [34,35,36]. Wheel positioning actuators and tyre pressure systems are extensively investigated [6,37,38,39,40], and after reaching a high enough maturity level for industrial application, they may be integrated into the longitudinal and lateral ICC systems. Investigations in a simulation environment demonstrated the potential of this solution [41,42]. The integration of drivetrain control (e.g., active differentials and electric motors) with active steering falls under this category and has undergone intensive investigation [43,44,45,46,47,48,49,50,51]. Few of those studies explicitly mentioned the incorporation of the combined slip effect in controller design. For instance, in [34], the authors employed input-output linearisation and sliding-mode control (SMC) to address the nonlinear Dugoff tyre model. Acarman [22] also adopted the Dugoff tyre model and utilised the state-dependent Riccati equation (SDRE) technique. Falcone et al. [21] explored optimal control input online with a four-wheel vehicle model incorporating wheel dynamics and the Pacejka tyre model. Researchers in [20,25] adopted a linear parameter-varying (LPV) model. Linear tyre models are frequently employed for simplification, assuming small slips in both directions. With a linear tyre model, the linear quadratic regulator (LQR) becomes a favourable choice for control strategy due to its optimality feature and has been adopted in several studies [5,16,24,44,49,51]. In [14,46], the authors utilised adaptive control to estimate cornering stiffness online. In most cases, this assumption is acceptable since the controller prevents the vehicle from entering a state where tyre force saturation can occur, and performance is satisfactory in simulated tests such as Double Lane Change and Sine-with-Dwell manoeuvres. 

The third category involves longitudinal and vertical system integration. Notably, the application of suspension and braking systems or drivetrain for yaw stabilisation has primarily been explored in earlier years. Smakman [52] integrated an active suspension with a brake-based ESC system, examining two integration approaches to mitigate disturbances between actuators and enhance overall system performance. The active suspension is predominantly employed as an active anti-roll bar, altering load transfer between the front and rear axles, with an extended function enabling the attainment of negative roll stiffness. Hac and Bodie [53] first calculated the capability of various actuators to generate the corrective yaw moment. The combination of active braking and magneto-rheological (MR) dampers was then adopted in the proposed ICC system, which used a supervisory level to decide the control authority of actuators. In [54], the authors attempted to design a global vehicle controller that coordinates independent brakes and active suspension to track a reference yaw rate. However, the authors reported the controllability problem during the transition phase of the turn. Consequently, the active suspension was only exploited for improving ride comfort, i.e., minimising pitch, roll, and vertical motion. It has been noticed that papers under this category commonly calculate look-up tables offline to control the suspension system, due to the high model complexity if vertical dynamics are included.

The fourth category Involves lateral and vertical system integration. The earliest exploration of coordinating active suspension and active steering was conducted by Yokoya et al. [55] during the development of a sports car at Toyota. The controller predominantly employs rule-based functions, utilising rear-wheel steering and active suspension to enhance stability. The rear steering angle is proportional to that in the front and can switch between reverse-phase, neutral, and same-phase. The switching is solely based on longitudinal velocity, i.e., reverse-phase for low velocity, same-phase for high velocity, and neutral for intermediate range (a concept still adopted by multiple manufacturers today). Active hydraulic suspension can adjust body height and roll stiffness distribution at high velocity. The integration of active suspension and active four-wheel steering for a μ-split braking manoeuvre was tested in [56], resulting in comprehensive improvements in yaw stabilisation and ride comfort. The controller was developed using a modified quarter-car model, incorporating vertical dynamics, wheel dynamics, and tyre forces at a single corner of the car. March and Shim [57] employed fuzzy logic to control active front steering and active suspension. The fuzzy controller’s output is a reference for the low-level normal force controller, based on a 2-D look-up table. In [58], the authors developed an integrated power steering system and active suspension. Alternatively, in [8], an active anti-roll bar was utilised instead of active suspension. Currently, controllable suspension has become widespread and may be installed in production vehicles starting from the C-segment, which has led to novel research in the field [4,59].

The fifth category involves full integration, combining longitudinal, lateral, and vertical domains. A few studies have undertaken comprehensive coordination of all three variables. Kawakami et al. [60] used rule-based coordination to integrate four-wheel steering, active suspension, traction control and anti-lock braking system (ABS). The study suggested that the active suspension (active roll moment distribution) is more effective in the high-acceleration range. At the same time, the four-wheel steering works better with low to intermediate lateral acceleration.

Control strategies are necessary for active vehicle systems. Rodic and Vukobratovic [61] proposed a synthesised control strategy with an advanced vision for an autopilot. The system aims to track a pre-programmed trajectory through the integrated control of four-wheel steering, active damping and independent wheel torque control. The controller was designed with knowledge of the vehicle dynamics and uses the proportional–integral–derivative (PID) technique at a low level. Simulation proved that the controller can accurately track the trajectory and is robust against disturbances like changes in friction, wind, etc., when lateral acceleration is low (up to 2.6 m/s^2^). In paper [3], a desired yaw moment is calculated with vehicle state information and driver input using PID control. The moment is distributed among the subsystems according to specific demands, e.g., comfort- or safety-orientated strategy. The distribution also considers whether one or two subsystems encountered a failure. The proposed system performs better than a conventional ESC system that relies only on differential braking. In [62], the authors used a desired yaw moment as the output of the high-level controller, employing an SMC technique for its calculation. The distribution of yaw moment is determined by the necessary condition for optimising a cost function, while roll moment distribution is controlled by a simple proportional-integral (PI) controller. The results showed that the contribution of the anti-roll bar to enhance yaw stability could be ignored. It is not surprising though, since the reduction of roll angle due to the active anti-roll bar took more than 1 s to become visible. This suggests that there may be a need for future efforts in this area. The constraint of friction limit is included in the optimisation of yaw moment distribution [63]. In this paper, steady-state load transfer is considered in coordinating active front wheel steering (AFS) and brake-based ESC. Yet the actual vertical load is influenced by the control of active dampers, which regulates roll moment distribution using SMC [64].

Vehicle electrification introduces new perspectives for ICC development. As reported in [65], EVs can exhibit over 20% higher mass than their internal combustion engine counterparts. Furthermore, integrating IWMs in EVs significantly increases the unsprung mass (UM), potentially impacting ride comfort and vehicle handling. In response, ICC designs for EVs may incorporate semi-active or active suspension systems with additional functionalities such as roll and pitch control [66] and ride blending (concurrent control of the action of IWMs and active suspension) [67].

Designing ICC for EVs requires careful consideration of the fact that the energy consumption of active subsystems may affect the overall energy efficiency, potentially reducing the vehicle’s mileage per charge [68]. Transitioning from high-level feedback controllers, which are common in many ICC variants [69], to feedforward solutions can mitigate the issue of increased energy consumption [70].

ICC is also crucial in enabling various Advanced Driver Assistance System (ADAS) functionalities. These include increased passenger comfort, redundant driving safety control, overall energy consumption reduction, specific tasks related to PT, decision-making, and environment perception. Numerous studies focus on enhancing AD PT performance [71,72,73,74,75], lateral stability [76,77,78,79], and energy consumption [80,81].

As AD technology advances, simultaneous consideration of multiple objectives becomes crucial, as they often interfere [82]. Addressing these challenges involves measures at the controller level logic, incorporating predictive components as demonstrated in [83,84,85].

Over the last 30 years, PT tasks have been extensively researched [86,87,88,89]. Most of them were developed for automated robots performing at low velocities and reached a high enough maturity level for industrial application. PT control approaches in the context of AD address issues related to parametric uncertainties, and external disturbances that cannot be avoided [90]. Current investigations are performed in the automotive field, where velocities are high, and low-friction roads and roads with irregularities are widespread [91,92].

Recent works have shifted focus towards ICC implementation in the realm of AD. For instance, in [93], the authors developed an AFS and braking ICC system designed for emergency collision avoidance in autonomous vehicles (AVs). Another study [94] delved into PT and ICC for obstacle avoidance, with a primary emphasis on improving PT [95,96].

Literature reviews published in recent years have explored ICC architectures for conventional vehicles [1,97,98,99,100]. Simultaneously, another group of review papers has examined PT tasks specifically in the context of AVs [101,102,103,104,105]. In both research and review scientific publications related to ICC and PT, these problems are investigated separately. In ICC-related studies, the vector of control inputs typically involves forces and moments. In contrast, in PT tasks, the steering angle emerges as the most frequently employed control input. Recognising steering as a potential tool for regulating lateral and yaw dynamics, the evolution of AD technology focusing on PT introduces opportunities for further advancements in ICC techniques. Simultaneously, integrating longitudinal, lateral, and vertical system dynamics offers additional prospects for enhancing PT performance. The incorporation of PT and ICC holds promise for synergetic development, where advancements in one domain can contribute to the refining techniques in the other.

## 3. Common Controller Layout for Automated Vehicles

The main integrated vehicle control structures include decentralised, centralised, and coordinated architectures presented in Figure 1. The decentralised architecture, sometimes referred to as downstream [106], is extensively utilised by original equipment manufacturers (OEMs). This preference arises from their ability to procure systems from suppliers and subcontractors (Tier 1 and Tier 2), who develop both software and hardware.

This approach minimises the need for extensive integration, mainly relying on network-based communication. Such systems are easy to fine-tune, but a key drawback is the potential for malfunctioning operations. In some instances, decentralised systems incorporate a coordination layer that collects outputs from individual controllers and returns corrected values of the same variables to multiple actuators [1]. While this coordination layer helps mitigate issues, it may not eliminate them. A notable example of a decentralised structure featuring a coordination layer is an ESC system produced by Tier 1.

The centralised architecture is sometimes referred to as the upstream architecture [106]. It solves the principal drawback associated with decentralised systems. In the centralised architecture, a high-level controller coordinates subsystems and prevents conflicts through control allocation (CA). The development of the master controller requires collaboration between OEMs, Tier 1, and Tier 2. Integrating new actuators into such a system after its development can be challenging due to an increase in computational power and the reluctance of Tier 1 and Tier 2 to share algorithms. 

The third category is the coordinated or multi-layer structure, which belongs to upstream architectures. It includes a high-level controller, where the required demand is calculated, middle-level control, where the control allocation task is solved by taking into account saturation and limitation of the actuators, and a low-level controller where separate actuators are controlled.

Using a decentralised control structure, it is impossible to realise ICC, as all the controllers work independently. The centralised control structure has only one controller and cannot practically conform to the existing vehicle control system development mode due to (i) lack of modularity, which requires the OEMs to develop the controller together with Tier 1 and Tier 2, (ii) complexity of the controller, (iii) lack of flexibility when additional actuators are needed, and (iv) system failure in case of controller breakdown [1,107]. This work focuses on coordinated architecture as one of the most perspective ones for ICC implementation. The schematical overall structure of the automated vehicle with integrated chassis control is presented in Figure 2 and contains sensing, data pre-processing, perception, planning, reference generator, and high-level/middle-level/low-level controllers.

In the schematic representation, the reference generator and middle-level controller contain several blocks with the designated green colour. This colour indicates that the saturation and constraints are applied to the reference values. Some of the constraints may be guided by the values provided in the standards to ensure safety and comfort, and others by physical limitations. The symbols in the scheme are explained briefly as follows. The high-level controller produces v—control demand vector, which is fed into the middle-level controller. In the CA part, required control output values are generated, based on the control demand vector, system constraints, and saturation. These are the following: $F_{zi}^{CA}—$vertical forces, $\delta_{i}^{CA}$—wheels steering/toe angles, $\gamma_{i}^{CA}—$wheels camber angles, $M_{em,\;i}^{CA}—$propulsion torques on wheels, and $M_{br,\;i}^{CA}—$braking torques on wheels. Based on these target values, individual low-level controllers generate the required voltage and current values.

For AD, many sensors (Figure 3) should not only provide information about the vehicle states but should also be combined with data processing to enable localisation, environment perception, and road conditions. With these data, the AV control algorithm could perform decision-making and motion planning similar to a human driver [108,109,110,111]. The main sensors available in AD, which can be used for ICC are presented in Figure 3. Using advanced sensors available in AD, the performance of ICC may be increased. Sensor systems can be used to acquire measurements about vehicle states, location, and environment.

Sensors can be categorised into internal and external state sensors, and passive and dynamic sensors [108]. Internal state sensors collect information about vehicle state, position, events, and changes. They include in-vehicle sensors and global navigation satellite systems (GNSSs). External state sensors collect information about the environment such as cameras, radio detection and ranging (radar), light detection and ranging (LIDAR), and ultrasonic sensors. These types of sensors can be categorised by transmission range into short, medium, and long-range [108]. Passive sensors collect external energy and output information such as vision cameras, inertial measurement units (IMUs), and GNSS, while dynamic sensors emit energy and collect responses from the environment such as radars and LIDARs [108]. It should be noted that sensors often require calibration and recalibration. Cameras require lens distortion calibration [112], while IMUs should be calibrated for temperature, bias- [113], etc. In addition to sensors, V2X communications potentially provide infrastructure-related and other information during driving, which can be very valuable and enable a high level of convenience that is not possible otherwise [114].

In-vehicle sensors are available in conventional vehicles, and they play a crucial role in measuring essential vehicle parameters. However, regarding vehicle demonstrators, additional sensors are frequently incorporated to implement control strategies. This scenario differs from production vehicles, where the utilisation of sensors is constrained due to their high cost. Traditional ICC depends heavily on human interface sensors for steering, acceleration, and deceleration to calculate the basis of reference values [115]. As for AD vehicles, global and local targets need to be set [116]. Therefore, the main sensors that can still be used in ICC are IMU [117], steering angle, angular wheel velocity sensors, tyre pressure measurement sensors (TPMS), vertical wheel displacement sensors, external temperature sensors, light and rain sensors, sensors for battery monitoring systems in EV such as current, voltage and temperature. IMU, angular wheel velocity sensors, and TPMS [118] can also provide useful real-time information for vehicle state estimation [119]. External temperature, rain, and light sensors can provide basic weather and road state information important in tyre friction estimation. Together with the camera, IMU, and angular wheel encoders, these sensors can provide information about road surface friction coefficient [120]. The functions, advantages, and disadvantages of the types of main sensors are summarised in Table 1.

GNSS provides positioning and navigation data to assist with route planning and vehicle positioning on the road [109,121]. The usual GNSS update rate is only 1–10 Hz. Nonetheless, when it is combined with IMU to achieve GNSS assisted inertial navigation system (GNSS/INS) capabilities, higher update rates of 400 Hz for position, velocity, and attitude and 800 Hz for pitch, yaw and roll angles, rates, and three-axis accelerations can be achieved. State-of-art tactical grade GNSS/INS modules achieve a horizontal positioning accuracy of 1 m without real-time-kinematics (RTK) [122]. Based on very useful positioning, heading, and dynamics information, this sensor is essential for AD [123]. However, it needs to be more robust to be used on its own due to the possibility of poor navigation signal quality in cities with dense and high buildings. Therefore, there should always be an alternative system to provide positioning redundancy on roads [124,125,126,127]. Such systems require a higher level of integration that is enabled by localisation methods in the perception layer.

The recent decade brought considerable advances in computer vision, making camera devices almost as valuable as human vision in driving [108,109]. Human and camera systems have common and separate advantages and disadvantages when used for driving (Table 1). The cameras provide a 2D or 3D visual perception of the environment, and thermal cameras provide thermal information that can be useful in bad weather conditions or at night to detect pedestrians, animals, and other vehicles. The main characteristics of such sensors are resolution, field of view (FoV), and light sensitivity. Currently, visual and thermal cameras are used in ADAS systems, which can also be used in monocular and stereo vision systems to provide information about the environment, road conditions, and previews [108]. 

The cameras are affordable and have quite good sensitivity during the night; yet, the data that they provide require pre-processing and complex perception algorithms. The advantages of vision cameras are cost-effectiveness, an abundance of information about the environment, and the availability of advanced processing algorithms, whereas the disadvantages are huge processing resources and lower or poor performance in low light and severe weather conditions. The advantages of thermal cameras are the ability to perform well at night and discern warm living objects from the environment. Their drawbacks are that they are more expensive, have low resolution, and are monochrome. Cameras can also work in stereo configurations to detect depth and distance from disparity. ICC cameras can be used for vehicle state estimation, environment, and other traffic participants’ detection.

LIDAR projects infrared light and senses reflections from objects to estimate the distance by measuring the time of flight and surface reflectivity through the relative amplitude of the reflected signal [108]. LIDAR technology is constantly improving and has recently become affordable as a standard vehicle sensor. Currently, there are mechanical spinning LIDARs, solid-state spinning LIDARs, and static solid-state lidars. Even with current prices, LIDARs are too expensive for use in production vehicles. LIDAR can be used in 2D and 3D localisation [121] and object detection and tracking [128]. Yet most of the LIDARs are bound to weather impact [129] and have a limited minimal range of 0.5 m and a maximal range of operation of 50–100 m. The main characteristics of these sensors are vertical channels/resolution, working distance range, point rate, and vertical and horizontal FoV. LIDARs produce point clouds, where points are reflection points in space and the value in this point shows the reflectivity of the point or intensity of reflection [108]. The processing of such 3D point clouds requires complex surface detection and object detection and tracking algorithms to make them useful. An example of such an algorithm could be HYDRO-3D, where the historical object tracking information is leveraged to assist the inference for object detection; such an approach improves object detection performance with short-term occlusion and out-of-range issues [130]. Usually, it is hard to place one LIDAR on the vehicle to cover 360 degrees horizontally as there cannot be any obstructions in FoV. Therefore, LIDARs often are mounted in the centre and are lifted from it to cover the space as close to the vehicle as possible, but there will still be some blind zones in the closest space. Other configurations include mounting two LIDARs on the front and back of the roof or mounting 180-degree horizontal FoV LIDARs on four sides of the vehicle to achieve the best coverage. Lidar data are suitable for accurate distance measurements, object detection, obstacle detection, localisation, and mapping. Usually, LIDARs are not used without combining them with cameras in the same AD vehicles [108,119].

Radar works by emitting electromagnetic waves and collecting the reflections [108]. State-of-the-art radars use frequency-modulated continuous wave ultrawide-band signals and measure the time of return, strength, and frequency Doppler shift between transmitted and received reflected signals. In the last two decades, radar has become inseparable from ADAS systems such as adaptive cruise control (ACC), front collision prevention, lane change assistance, and AD in highway traffic jams [108,121]. In contrast to cameras, radars measure the distance to objects and relative velocity to the radar. Modern radars even support radar imaging using multiple-input multiple-output (MIMO) grid antennas. This allows for higher resolution and multi-object detection and tracking using radar in various weather conditions [131]. Radars are used for distance measurement, velocity measurement, and object detection in various weather and illumination conditions. The advantages are affordable price, a robust basic principle of work, the ability to detect vehicles, humans, and animals, possibility to be mounted behind plastic surfaces such as bumpers, while the disadvantages are that they only measure a single area or have poor resolution, the processing of radar signals can be complex due to multi-path reflections, they do not see stationary objects well, and they do not see objects that do not reflect the microwave frequencies. Such sensors, together with cameras, can improve object tracking precision and trajectory estimation. Radar data can be combined with video data in data-pre-processing and perception to achieve the most valuable results. Usually, there are multiple radars in one AD vehicle with short- and long-range coverage; short-range coverage provides an understanding of the close environment and lane change, and long-range coverage is for ACC and emergency obstacle detection.

Ultrasonic sensors are currently widely used in ADAS such as parking assist systems and obstacle proximity warning systems and as redundant or cheaper sensors in lane change assistants [132]. These sensors emit ultrasound chirps and detect objects and their distance to them based on the time of flight of reflected sound waves. Ultrasonic sensors work well in various weather conditions, except for sensor contamination. Their advantages are simple working principles, low price, fast processing, and functionality in various illumination and weather conditions if it is not partly covered. The disadvantages are that they measure the distance to the closest object in the ultrasonic beam and the distance measurement is unstable and may vary a lot; therefore, advanced processing is required to obtain stable measurements [132]. These sensors can also be replaced or supported by short-range radars, infrared proximity sensors [133], or single-channel LIDARs. Yet, optical sensors are always more susceptible to weather particles and dust.

In the near future, AVs may use cooperative sensing that shares information through vehicle-to-vehicle (V2V) and vehicle-to-infrastructure (V2I) communication [134]. V2I can provide information about the state of intersections and traffic lights, speed limits, and road hazards that may be used in ICC to optimise velocity to go on a green light at the intersection, to adapt the suspension to road roughness, choose a safe and comfortable velocity and be alert for unusual situations on the road. Also, such communication will enable sharing an accumulated experience about road roughness in the cloud so that each vehicle ICC can prepare vehicle systems in advance for changes in road conditions based on GNSS-based or localisation algorithms-based locations [114,135]. V2V communications will allow the sharing of real-time information about AD vehicle planned maneuverers and trajectories to provide smoother driving comfort [111,136,137,138,139]. The advantage of vehicle-to-everything (V2X) communication systems is access to broad information from V2I and other V2V and access to computing and data resources in the vehicle-to-cloud (V2C) systems; the disadvantages of such systems are strict latencies for real-time operations [140] and cybersecurity [141]. Cyberattacks could create serious safety issues and lead to accidents, posing a physical threat to users or passengers [142]. Due to cybersecurity and the possibility of losing the connection, AD and ICC should always have fallback solutions, or use V2X as supplementary information to already working self-sufficient sensor systems. Reliance on cooperative sensing will reduce the demand for and the total cost of onboard sensors, making this approach highly cost-effective [143].

Sensor data pre-processing is now essential [110]. By combining sensors using kinematic and dynamic models in Kalman Filters and other adaptive filtering methods, the data from several sensors can be fused using the sensor fusion approach to achieve robustness, stability, and accuracy [144,145,146]. On the other hand, the data pre-processing step can involve virtual sensors that add additional measurements that cannot be provided by physical sensors or that are not available or costly to integrate. Additional sensor pre-processing involves deep learning [109]. The pre-processing step should also include methods for improving fault tolerance, and observers can be used for sensor fault estimation [147]. Kalman filters estimate different parameters using data from low-cost sensors [95].

Perception is an essential part of various AD and ICC systems [109,148]. Perception can include understanding and prediction of the local environment, road situation, and vehicle states. The perception methods such as object classification, object tracking, and vehicle localisations make sense of raw data that are brought after the pre-processing step by sensors and V2X communication. AD and ICC can use cameras to estimate velocity using visual odometry, yaw, roll, and pitch rates, detect road lines [149,150], road curve angle and road bank angle, intersections, railroad, and pedestrian crossings, road surface type and conditions [118], road lanes and boundaries [151,152], road signs including warnings and speed limits [153], and horizontal marking detection and recognition [154], road damage, potholes, and distress detection [155], location [156], velocity and displacement of the vehicle [157] and other vehicles on the road even using their taillights [158]. ICC systems can use previously described advanced sensors to acquire horizon prediction, and longitudinal and lateral road surface slopes, which, together with in-vehicle IMU and GNSS/INS, will enable a more accurate decomposition of linear and gravitation-caused acceleration. The camera, combined with LIDARs, can provide valuable 3D information about surface geometry in front of the vehicle to the nearest 50–100 m, as well as information about objects and the distance to them in higher angular resolution than radar technology. Object detection and tracking algorithms can rely on LIDAR, radar, and camera data to acquire environmental and situational awareness.

Vehicle localisation is highly relevant for global and local planning. It can be implemented using direct positioning using GNSS and adding accuracy using LIDARs and cameras with simultaneous localisation and mapping (SLAM), visual odometry, and map matching methods [109,159,160]. Close surroundings and situational awareness based on radars, LIDARs, cameras, and ultrasonic sensors are used for trajectory planning [109]. In addition, the navigation system cannot provide much information about the road surface and curvature that would be essential for ICC. Thus it will be used together with V2X communication to cloud databases that are built by vehicles driving on the roads and sending telemetry to get actual data about the road [161,162]. Another way of getting the road surface information is by having detailed road profile maps saved in the vehicle’s memory and by periodically updating it This way, GNSS/INS with V2X communication enables an experience-based preview of the road surface in addition to navigation and road maps, which helps ICC to prepare for lateral accelerations on road curves, altitude changes, and decelerations and accelerations on traffic lights and intersections in advance [134,163,164]. Vehicle states such as longitudinal, lateral and vertical velocities, heading, and pitch and roll angles can be estimated by the fusion of data from various sensors and systems that overlap and therefore provide robustness and additional accuracy.

Planning covers the AD decision-making processes from trajectory and behaviour generation in the moment to global decisions on the overall aim of the movement, such as routing from start to destination point [109], and responding to complex manoeuvres such as roundabout merging [121]. It involves global planning, behavioural planning, local trajectory generation and trajectory validation.

Global planning considers road networks, traffic conditions, and any known obstacles or restrictions [165,166,167]. It sets long-term goals and paths for the vehicle to follow and goals to achieve. Global planning re-assembles human route planning using detailed 2D and 3D maps [121], navigation, and personal and publicly available information. Based on the fact that global planning is performed before starting the ride, and during any changes in the situation or objectives of the ride, it can even be offloaded to the cloud, and it can work using V2C communication. To improve at least basic fallback in case of broken connection and no network, the systems should provide at least limited global planning. For global planning, the system should be provided with the current global position of the vehicle, all destination points and desired arrival times if possible, a range that can be covered with a current charge or fuel amount, and the time when it will plan and solve any conflicts related to the objectives and add any required stops to refuel. Also, it sets constraints on lower planning levels such as max velocity, comfort level, etc. This planning would depend on vehicle localisation and V2X communications.

Behavioural planning occurs between the start and finish of the ride. It involves more immediate decisions, such as when to change lanes, selection of safe target velocity in current road conditions and before the upcoming road curves, or navigating through a busy intersection [167,168,169]. Behavioural planning works with constraints set by the global plan but reacts to the nearby environment and situations that happen in real time and are predicted for the next moments. It may optimise the driving comfort, arrival duration, energy efficiency, safety, wear of components, etc. Behavioural planning would greatly depend on the perception of environmental awareness enabled by object detection, classification, vehicle localisation, object tracking, and trajectory prediction, which relies on sensors and V2X communication [136]. It takes the local real-time situation of road surface and road obstacles to make decisions.

Trajectory generation or local planning is the creation of a reference path that the vehicle will follow, defined in terms of waypoints, velocity, and acceleration over time, known as PT in the literature [170,171]. It may also be bound by references for longitudinal velocity, lateral and longitudinal accelerations, and jerk set by global/behavioural planning. The local surroundings’ understanding and prediction together with vehicle dynamics is essential in this process. The trajectory should be aligned with constraints imposed by global planning for improving comfort and stability, for example, by limiting accelerations and reducing jerk. At the same time, trajectory generation may account for unexpected obstacles found during the ride [94]. Also, trajectory generation and PT are key to safe lane changes [139]. For a safe trajectory, the information can be accumulated from vehicle sensors and combined with other vehicle information, including current state and planned trajectories and behaviour using V2V communication [94].

After a detailed exploration of sensing technologies and their role in AD with ICC, it is essential to understand their integration with controllers and actuators, as all these elements form the backbone of vehicles’ dynamic control systems. In conventional vehicles, the driver directly impacts three primary parameters: acceleration, braking, and steering [172]. In AD, the fundamental situation remains the same, but reference parameters depend on the environmental perception and decision-making systems. As the next step, reference values are compared to the measured or estimated ones, and a high-level control strategy is implemented for defining control demands. High-level controllers may range from extremely simple, such as PID, gain-scheduling controllers to highly complex, such as adaptive fuzzy SMC, nonlinear model predictive control (NMPC) [173,174,175]. Traditionally, feedback control was employed for such tasks [176]. However, with the advancement of AD technologies, particularly in artificial intelligence and the focus on perception tasks, feedforward and control with preview can now be realised. This way, AD vehicles can plan and distribute control actions in advance, such as the preparation of the suspension actuators based on upcoming road irregularities [177], predictive pedestrian avoidance, [178] and path planning.

Since multiple actuators are available for control, which is expected to be the case in AD, the control allocator has to provide a way to allocate the control effort appropriately. It is conducted on a middle layer, where the CA task is resolved, making it an effective way to implement fault-tolerant and energy-efficient control [2]. The middle level produces target values for the low-level controllers, additionally limiting actuator targets when the control allocator requests more control effort than the system is physically capable of producing, a so-called saturation task [38,179].

The final component is the low-level control; commonly, the developer of actuators proposes both hardware and software. Therefore, their inner dynamics may be uncertain for the automotive manufacturer, and integrating several systems requires additional testing [180].

In the sections below, a detailed description of each of the control subsystems is provided.

## 4. Applied Control Methods

In this section, vehicle state estimators are presented. After that, the evaluation of reference parameters considering saturation is described. Finally, control methods used for high-level and middle-level controllers are presented.

### 4.1. Vehicle State Estimation

Two types of input parameters are used in control tasks: measured and estimated [181]. Due to the sensors’ price, accuracy, and packaging issues, not all of the parameters were measured directly on production vehicles. The main measurable parameters were accelerations in X, Y, and Z directions and angular rates of pitch, roll, and yaw. This information was received from the IMU. Other measurable parameters were the steering angle and angular velocities of four wheels [182]. In vehicles with controllable suspensions, additional displacement and (or) acceleration sensors were installed on UMs [145], and in some industrial applications, additional acceleration sensors were installed above absorbers on sprung mass (SM) [183].

Other parameters which are needed for vehicle control were estimated. These parameters were longitudinal velocity, vehicle body sideslip angle, cornering stiffness, tyre loads, vertical velocities of SM and UM, and other parameters needed for PT. 

There were no sensors for the measurement of longitudinal velocity in production vehicles. Longitudinal acceleration and angular wheel speed, which were both measured, can be used for estimation. Theoretically, longitudinal velocity can be estimated using the formula presented below [184]:(1)$$V_{simple}=r_{ef}\omega +\sum_{i=k_{no\;brake}}^{k}a_{x,meas}\left(i\right)dt,$$
where $r_{ef}$ is the dynamic tyre radius, ω is the angular velocity of the selected wheel, $a_{x,meas}$ is the measured acceleration in the longitudinal direction, and dt is the time step. Such an approach is not accurate as there are several sources of errors. First, errors occur during the estimation of the effective radius. Second, there is noise from angular velocity and acceleration sensors. Biases can occur frequently while acceleration is integrated over time dt (second term in Formula (1)). To address these issues, the Kalman filter is commonly adopted. Longitudinal velocity can be estimated using the formula [184]:(2)$$\hat{V}\left(k\right)=\hat{V}\left(k-1\right)+a_{x,meas}\left(k\right)dt+K_{1}\left(\omega \left(k\right)-\frac{1}{r_{est}}\left(\hat{V}\left(k-1\right)+a_{x,meas}\left(k\right)dt\right)\right),$$
where $K_{1}$ is the Kalman gain, $r_{est}$ is the estimated tyre radius, and k is the sample step.

Body sideslip angle can be measured using optical flow or GPS sensors. However, this approach still has practical issues related to cost, accuracy, and reliability, which affect its use in production vehicles. The body sideslip angle depends on vehicle lateral and longitudinal velocities (Figure 4), and theoretically can be calculated using the formula:(3)$$\beta =tan^{-1}\frac{V_{y}}{V_{x}},$$
where $V_{y}$ is the lateral velocity, $V_{x}$ is the longitudinal velocity. 

The sideslip rate can be estimated without the need to use lateral velocity [185]:(4)$$\dot{\hat{\beta }}\left(k\right)\approx \frac{a_{y,meas}}{\hat{V}\left(k\right)}-{\dot{\psi }}_{meas},$$
where $a_{y,meas}$ is the measured lateral acceleration and ${\dot{\psi }}_{meas}$ is the measured yaw rate.

Therefore, the sideslip can be estimated using the following formula [185]: (5)$$\hat{\beta }\left(k\right)=\int \left(\frac{a_{y,meas}}{\hat{V}\left(k\right)}-{\dot{\psi }}_{meas}\right)dt.$$

The bicycle model is widely used for different control tasks due to its simplicity. The disadvantages of this model are that the mass, inertia, centre of gravity, and cornering stiffness of the tyres do not change due to load conditions, vehicle velocity, and road type, and this is not accurate in real operational conditions [186]. We discuss how to overcome the limitations of the bicycle model in the other sections.

During estimator development, it should be noted that measured acceleration in longitudinal and lateral directions may be affected by Earth’s gravity while driving on a road with a slope or bank [185].

In practical applications, sometimes, both measured and modelled acceleration are used. Bias can occur during the integration procedure shown in (5); therefore, to eliminate the bias, an additional term, based on measured and modelled acceleration, is added to the formula. Modelled acceleration can be achieved from tyre forces, as described in [69]:(6)$$a_{y,model}=\frac{F_{y,fr}+F_{y,r}}{m},$$
where $F_{y,fr}$ and $F_{y,r}$ are lateral forces at the front and rear axles of the vehicle, which depend on tyre slip, yaw rate, velocity, and friction, and m is the vehicle mass. There is external wheel force measurement equipment from Kistler and other manufacturers, which can be placed on vehicle demonstrators. There is a solution when the hub bearing is equipped with sensors to measure forces [187]. However, the solution is not commercially available in production vehicles yet. As a result, lateral and longitudinal forces were estimated using tyre models such as Brush, Dugoff, Pacejka, and others suitable for real-time applications. The drawback of such an approach is the risk of error occurring when using incorrect tyre/road parameters. However, it is not unlikely to occur if the tyre was parametrised correctly [181].

Currently, the most widespread estimator for sideslip angle is one based on the Extended Kalman Filter (EKF) [186]:(7)$$\hat{\beta }\left(k\right)=\frac{{\hat{\beta }}_{k}^{-}+\beta_{k}^{v}}{2},$$
where ${\hat{\beta }}_{k}^{-}$ is the predicted sideslip angle; the sideslip angle $\beta_{k}^{v}$ can be calculated as follows:(8)$$\beta_{k}^{v}=\frac{v_{STM}-h\left({\hat{\beta }}_{k}^{-}\right)}{H}+{\hat{\beta }}_{k}^{-},$$
where $v_{STM}$ is single track model wheel velocity without longitudinal slip, $h\left({\hat{\beta }}_{k}^{-}\right)$ is the predicted single track model wheel velocity, H is the linearised output, calculated as follows $H=\frac{\partial v_{STM}}{\;\partial {\hat{\beta }}_{k}^{-}}$. The sideslip angle defined in (8) is the estimated value and will be marked as $\beta_{est}$ in formulas below.

EKF provides robust results for sideslip angle on high and low friction roads, taking into account road slope and banked corners.

During the evaluation of cornering stiffness adaptive tyre-force model may be used [188]:(9)$$F_{y,i}=(C_{i}+\Delta C_{i})\alpha_{i},$$
where i represents the front or rear axle, $C_{i}$ is cornering stiffness, $\Delta C_{i}$ is adaptive cornering stiffness, and $\alpha_{i}$ is front/rear sideslip angle. The parameter $\Delta C_{i}$ was used to take into account changes in cornering stiffness. It was included in the state vector of the EKF, which was constructed with state, input, and measurement. More details can be found in [188]. Cornering stiffness estimation using EKF provides accurate and robust results [185,186]. The detailed estimation algorithm for cornering stiffness is presented in [181].

One of the primary challenges faced by these estimators is their limited accuracy when dealing with high wheel slips on low-friction roads or sudden changes in friction coefficients, often referred to as “friction jumps” [189]. An effective estimator example is the one based on the Unscented Kalman Filter (UKF); while demanding more computational resources compared to the EKF, it offers the advantage of not requiring the linearisation of the model [190]. Recent research in the field of Kalman filter development for nonlinear systems has introduced the concept of adaptive covariance matrices; this approach yields superior results, particularly in complex driving conditions.

In the last few years, so-called data-driven virtual sensors for vehicle state estimation have been proposed for application in the automotive industry. These data-driven approaches have the potential to replace model-based methods [145,191,192]. Data-driven estimators employ artificial neural networks (ANN), which necessitate datasets for training, testing, and validation, but do not rely on mathematical models and lead to higher accuracy. In essence, when developing a data-driven sensor, experimental data are required for each unique application. Additionally, a hybrid approach for data-driven estimators has been proposed [193], where a model-based approach is used together with the ANN, enhancing the robustness of the estimator. The hybrid approach is more accurate than purely model-based or data-driven approaches [194].

To sum it up, vehicle state estimation is a crucial task, especially with the advancement of AD technology. The number of sensors (provided in Table 1) will continue to rise, including cameras, radars, LIDARs, and other distance sensors becoming mandatory components for this technology. Simultaneously, estimators will undergo further development, taking advantage of the increasing computational power of AVs. At the same time, OEMs will increasingly explore the possibility of transitioning from physical sensors to virtual ones.

### 4.2. Reference Generator and Saturation

When vehicle parameters are measured/estimated they are compared to the reference values to realise feedback control strategy. The reference parameters needed for global control realisation are presented below. When the reference value reaches a physical limit, a saturation effect takes place. Commonly, the implementation of saturation is performed in the middle-level controller, for a better understanding by readers we will present it in this subsection.

Some of the reference parameters can be predefined by the driver/occupant, while others are dependent on driving conditions and require mathematical models. For example, in ESC systems, yaw rate is often used, and reference yaw rate is calculated using a bicycle model with the input of steering angle and vehicle velocity [38]. However, simple linear models can be used only for linear regimes of motion, on high-friction roads with accelerations below 5 [m/s^2^] [1,195], and with velocities below 40 [km/h]. The main advantage of such a solution is its simplicity. The situation is different when driving at high velocities or on low-friction pavements. In such a case, tyre dynamics need to be taken into account, which means that sideslip angle and tyre cornering stiffnesses need to be estimated as shown in Section 4.1.

Reference longitudinal velocity is used for several systems in vehicles, such as ACC and lane-keeping assist systems (LKAS). The minimal value $V_{x,min}$ for the ACC system is 5 [m/s] [196] and 20 [m/s] for LKAS. With an activated LKAS system, longitudinal velocity should not decrease more than 5 [m/s] during manoeuvre [197].

For the vehicle cruise control system, reference longitudinal velocity is needed, and it can be set as constant by the driver/occupant. For comfort improvement using semi-active suspension, vertical acceleration and vertical velocity of SM can be used as one of the metrics, and the reference can be set to 0.

Reference longitudinal acceleration is used for vehicle control as well. When the reference value reaches a physical limit or one defined by system developers, a saturation effect occurs. The realisation of saturation is described below:(10)$$a_{x,sat}=\left\{\begin{array}{c} a_{x},\left|a_{x}\right|\leq a_{x,max}\\ \pm a_{x,max},\left|a_{x}\right|>a_{x,max} \end{array},\right.$$
where $a_{x,max}$ is the maximal value of longitudinal acceleration that can be achieved from the tyre friction ellipse:(11)$$a_{x,max}=\sqrt{\mu g^{2}-a_{y}^{2}},$$
where μ is the maximal friction coefficient and g is the gravitational acceleration.

To implement this, we need to define maximal road friction. It can be achieved during braking (more details can be found in [198]).

Longitudinal acceleration can be additionally limited to improve ride comfort. For example, in ACC systems, acceleration is limited during braking at velocities higher than 20 m/s, with $a_{x,max}=-3.5\;\left[m/s^{2}\right]$,. After that, $a_{x,max}$ linearly decreases with a decrease in velocity and reaches $-5\;\left[m/s^{2}\right]$ at velocities lower than 5 m/s. At velocities higher than 20 [m/s], maximal positive acceleration is limited to $2\;\left[m/s^{2}\right]$ and is linearly increasing with a decrease in velocity before it reaches $4\;\left[m/s^{2}\right]$ at velocities lower than 5 [m/s] [196].

The saturated steady-state response (10) cannot describe the dynamic behaviour of the vehicle. Therefore, the second-order transfer function can be used for this purpose, and the final longitudinal acceleration reference value is set to be:(12)$$a_{x,reff}=\frac{w_{k}^{2}\left(1+\tau s\right)}{s^{2}+2\xi w_{k}s+w_{k}^{2}}a_{x,sat},$$
where $w_{k}$ is the natural frequency and ξ is the damping ratio. More details about these parameters can be found in [1].

Reference lateral acceleration can be calculated using Formula (6) or from the bicycle model [199]:(13)$$a_{y}=\frac{V_{x}^{2}\delta }{L},$$
where δ is the steering angle and L is the wheelbase. Lateral acceleration has limits that are defined by:(14)$$a_{y,sat}=\left\{\begin{array}{c} a_{y},\left|a_{y}\right|\leq a_{y,max}\\ \pm a_{y,max},\left|a_{y}\right|>a_{y,max} \end{array}.\right.$$

From tyre friction ellipse:(15)$$a_{y,max}=\sqrt{\mu g^{2}-a_{x}^{2}}.$$

Sometimes $a_{y,max}$ is additionally limited. For example, in LKASs, lateral acceleration is limited to $a_{y,max}\leq 3\left[m/s^{2}\right]$ [197]. The second-order transfer function can be used to repeat the dynamic behaviour of the vehicle in a similar way as was shown in Formula (12).

Reference yaw rate is commonly calculated using a bicycle model (Figure 4). The reference yaw rate for a steady-state case of a four-wheel steering vehicle is presented below:(16)$${\dot{\psi }}_{reff,ss}=\frac{V}{L+K_{us}V^{2}}(\delta_{f}-\delta_{r}),$$
where $K_{us}$ is the understeer gradient, V is the vehicle velocity, and $\delta_{f}\;\mathrm{and}\;\delta_{r}$ are the front and rear axle steering angles, respectively.

The estimated (available) friction needed to constrain the desired steady-state response can be selected based on the available friction. The maximum reference yaw rate can be determined as follows [38]:(17)$${\dot{\psi }}_{max}\approx 0.85\frac{\mu g}{V_{x}}.$$

Maximal friction coefficient can be estimated through different techniques, for example, by using ANNs and computer vision, as shown in [120]. The saturated reference yaw rate is described as follows:(18)$${\dot{\psi }}_{sat}=\left\{\begin{array}{c} {\dot{\psi }}_{ref,ss},\;\left|{\dot{\psi }}_{ref,ss}\right|\leq {\dot{\psi }}_{max}\\ \pm {\dot{\psi }}_{max},\left|{\dot{\psi }}_{ref,ss}\right|>{\dot{\psi }}_{max} \end{array}\right..$$

The second-order transfer function can be used to repeat the dynamic behaviour of the vehicle in a similar way as was shown in Formula (12).

Reference sideslip angle $\beta_{ref}$ can be set to some threshold with an upper limit. There are two approaches commonly used for estimation of limit values. The first one is velocity-independent [1,200]:(19)$$\beta_{limit}=tan^{-1}\left(0.02\mu g\right)\approx 0.02\mu g\;\left[\mathrm{rad}\right].$$

Another approach is to use a velocity-dependent value. The sideslip angle may not as important for small velocities such that Formula (19) may be changed to:(20)$$\beta_{limit\_2}=10{}^{\circ}-7{}^{\circ}\frac{V^{2}}{{\left(40\frac{m}{s}\right)}^{2}}.$$

Therefore, the reference value is recalculated as:(21)$$\beta_{ref}=\left\{\begin{array}{c} \beta_{est},\left|\beta_{est}\right|\leq \beta_{limit}\\ \pm \beta_{limit\_2},\left|\beta_{est}\right|>\beta_{limit} \end{array}\right..$$

Phase-plane $\beta -\dot{\beta }$ is also widely used for ESC systems. In the first stages, the stability region is defined for a vehicle, after which control is realised to keep the vehicle stable. Using phase plane, the reference sideslip can be calculated as follows [201]:(22)$$\beta_{ref}=\{\begin{array}{c} \beta_{est}-\beta_{limit}\left(1-\frac{{\dot{\beta }}_{est}}{{\dot{\beta }}_{limit}}\right),\beta_{est}\geq 0\wedge {\dot{\beta }}_{est}\geq 0\\ \beta_{est}+\beta_{limit}\left(1+\frac{{\dot{\beta }}_{est}}{{\dot{\beta }}_{limit}}\right),\beta_{est}<0\wedge {\dot{\beta }}_{est}<0\\ \beta_{est}-\beta_{limit}\left(1+\frac{{\dot{\beta }}_{est}}{{\dot{\beta }}_{limit}}\right),\beta_{est}\geq 0\wedge {\dot{\beta }}_{est}<0\\ \beta_{est}+\beta_{limit}\left(1-\frac{{\dot{\beta }}_{est}}{{\dot{\beta }}_{limit}}\right),otherwise \end{array}.$$

Reference friction is another parameter commonly used in vehicle control systems. Theoretically, friction coefficient is the ratio between the longitudinal/lateral force and the vertical one [202]:(23)$$\mu_{x}=\frac{F_{x}}{F_{z}}=\frac{a_{x}}{g},$$
(24)$$\mu_{y}=\frac{F_{y}}{F_{z}}=\frac{a_{y}}{g}.$$

Equations (23) and (24) furnish values of current friction, which may not necessarily represent the maximum. Maximal values can be determined by examining the derivative of force (or acceleration); when the derivative equals zero, the maximum value is attained.

Another very important parameter is the reference longitudinal tyre slip. The actual tyre slip is calculated using the formula:(25)$$\lambda_{i}=\left\{\begin{array}{c} \frac{V_{x}-r_{ef,i}\omega_{i}}{V_{x}},\;for\;braking\\ \frac{r_{ef,i}\omega_{i}-V_{x}}{r_{ef,i}\omega_{i}},\;for\;acceleration \end{array}\right.,$$
where $r_{ef,i}$ is the dynamic radius of the wheel, i=1…4, and $\omega_{i}$ is an angular velocity of an *i*-th wheel.

In the first approach, the reference wheel slip may be considered to be the constant. However, this approach may lead to suboptimal braking performance on certain surfaces. Another approach is to estimate the friction and then match it with the closest value from a predefined lookup table, available at [203], to identify the road type and subsequently select the appropriate reference slip. Implementing this approach can be challenging due to the presence of noise in the measured data. To overcome this challenge, one effective method involves using ANNs to classify the road type, as demonstrated in the research [120]. Once the road type is determined, the reference slip values can be selected from the lookup table. This approach is particularly valuable, as it allows predefining the reference slip before braking. However, it should be noted that reference slip values may vary for different tyres and pavement types. A highly promising approach for estimating reference wheel slip involves utilising a polynomial fitting algorithm that considers changes in longitudinal wheel force, as outlined in the research available at [204], where the reference wheel slip is achieved when a change of longitudinal wheel force becomes equal to zero. Notably, this method offers the advantage of producing reference values that are insensitive to variations in tyre and pavement types. There are a few more papers where reference slip is defined; however, they require additional sensors [205] and are not used in production vehicles.

Using active anti-roll bars, active dampers or air springs, the roll angle can be controlled as well. The reference roll may be set up to zero. In such a case for the region where actuators provide enough force actual roll will be similar to zero, due to tyre deformations these values will be slightly different. In other cases, roll acceleration may be used as reference, and can be set up to zero. Another approach is to have a tilting effect when the vehicle is oriented towards the corner to increase the comfort level. In such a case reference roll may be calculated as follows [206]:(26)$$\phi_{ref}=\left\{\begin{array}{c} k_{1}\sin\left(k_{2}a_{y}\right),\left|k_{2}a_{y}\right|\leq \pi /2\\ k_{1},\left|k_{2}a_{y}\right|>\pi /2 \end{array}\right.,$$
where $k_{1}$ is the maximum permissible roll angle, $a_{y}$ is the lateral acceleration calculated from (6) or (13), and $k_{2}$ according to [206] can be calculated as follows:(27)$$k_{2}=\frac{1}{k_{1}g}.$$

The reference values mentioned earlier are typically employed in production vehicles.

The primary objectives in realising AD include desired velocity keeping, lane keeping/changing, following another vehicle, and braking. The first task requires in-vehicle sensors, and for the last three, camera and radar data are required. The distance between the vehicle’s centre of gravity (CoG) and the lane’s centreline (referred to as $e_{y}$ Figure 4) requires tracking. Standard [196] stipulates that the vehicle’s longitudinal centreline relative to the target vehicle’s longitudinal centreline should be maintained within a threshold of 0.5 meters. The orientation (heading) error of the vehicle concerning the road (denoted as $e_{\psi }$ in Figure 4) is the next parameter which is tracked. Reference values of these two parameters should be set to minimal values. These parameters are controlled using feedback controllers. Simultaneously, feedforward control mechanisms must be implemented to account for road curvature–R. The reference value is estimated from visual data at a look-ahead distance $x_{LA}\;$and virtual look-ahead distance $d_{LA}$ as shown in Figure 4.

### 4.3. High-Level Controller

A high-level control algorithm is designed to compute a vector of virtual inputs to the ICC and PT systems, which may include brakes, electric motors, steering system, suspension, active anti-roll bars, wheel positioning, and dynamic tyre pressure system.

The virtual inputs are usually chosen as longitudinal, lateral, and vertical forces, together with yaw, pitch and roll moments, and angles including steering, camber, and toe, that equal the number of degrees of freedom that the motion control system wants to control [207]. The equations of motion can be written as follows [202]:(28)$$m\left(\dot{V_{x}}+\dot{\theta }V_{z}-\dot{\psi }V_{y}\right)\approx m\left(\dot{V_{x}}-\dot{\psi }V_{y}\right)=F_{x}=\left(F_{x,fl}+F_{x,fr}\right)\cos\delta_{f}-\left(F_{y,fl}+\;F_{y,fr}\right)\sin\delta_{f}+\left(F_{x,rl}+F_{x,rr}\right)\cos\delta_{r}-\left(F_{y,rl}+F_{y,rr}\right)\sin\delta_{r}-\frac{1}{2}\rho C_{x}A_{x}V_{x,res}^{2}-\varphi F_{z},$$
(29)$$m\left(\dot{V_{y}}+\dot{\psi }V_{x}-\dot{\phi }V_{z}\right)\approx m\left(\dot{V_{y}}+\dot{\psi }V_{x}\right)=F_{y}=\left(F_{y,fl}+F_{y,fr}\right)\cos\delta_{f}+\left(F_{x,fl}+\;F_{x,fr}\right)\sin\delta_{f}+\left(F_{y,rl}+F_{y,rr}\right)\cos\delta_{r}+\left(F_{x,rl}+F_{x,rr}\right)\sin\delta_{r}-\frac{1}{2}\rho C_{y}A_{y}V_{y,res}^{2},$$
(30)$$m\left(\dot{V_{z}}-\dot{\theta }V_{x}+\dot{\phi }V_{y}\right)\approx m\dot{V_{z}}=F_{z}=F_{z,fl}+F_{z,fr}+F_{z,rl}+F_{z,rr},$$
(31)$$I_{x}\ddot{\phi }=M_{x}+\dot{\theta }\dot{\psi }\left(I_{y}-I_{z}\right)\approx M_{x}=\left(F_{z,fl}-F_{z,fr}\right)\frac{b}{2}+\left(F_{z,rl}-F_{z,rr}\right)\frac{b}{2}+\;m_{spr}a_{y}\left(h_{cg}-h_{r}\right),$$
(32)$$I_{y}\ddot{\theta }=M_{y}+\dot{\phi }\dot{\psi }\left(I_{z}-I_{x}\right)\approx M_{y}=\left(F_{z,rl}+F_{z,rr}\right)l_{r}-\left(F_{z,fl}+F_{z,fr}\right)l_{f}-\;m_{spr}a_{x}\left(h_{cg}-h_{p}\right)+\left(h_{cg}-h_{d}\right)\frac{1}{2}\rho C_{x}A_{x}V_{x,res}^{2},$$
(33)$$I_{z}\ddot{\psi }=M_{z}+\dot{\theta }\dot{\phi }\left(I_{x}-I_{y}\right)\approx M_{z}=\left[\left((F_{y,fl}+F_{y,fr}\right)cos\delta_{f}+(F_{x,fl}+F_{x,fr}\right)sin\delta_{f}]l_{f}-\;\left[(F_{y,rl}+F_{y,rr}\right)cos\delta_{r}+(F_{x,rl}+F_{x,rr})sin\delta_{r}]l_{r}+[(F_{x,fr}-F_{x,fl})cos\delta_{f}-(F_{x,fr}-\;F_{x,fl})sin\delta_{f}]\frac{b}{2}+[(F_{x,rr}-F_{x,rl})cos\delta_{f}-(F_{x,rr}-F_{x,rl})sin\delta_{f}]\frac{b}{2}=F_{y,f}l_{f}-F_{y,r}l_{r}+\;(F_{x,fr}-F_{x,fl})\frac{b}{2}+(F_{x,rr}-F_{x,rl})\frac{b}{2},$$
where $F_{x,i}$ is the longitudinal tyre force, i=fl,fr,rl,rr–front left, front right, rear left and rear right wheels, $F_{y,i}$ is the lateral tyre force, $F_{z,i}\;$is the vertical tyre force, $C_{x}$ is the frontal drag coefficient.

$A_{x}$ is the frontal cross-section of the vehicle, $V_{x,\;res}$ is the resulting in longitudinal velocity taking into account wind velocity, φ is the rolling resistance of the tyre, $C_{y}$ is the side drag coefficient, $A_{y}$ is the lateral cross-section of the vehicle, $V_{y,\;res}$ is the resulting in lateral velocity taking into account wind velocity, $\dot{\psi }$ is the yaw rate, $\dot{\theta }$ is the pitch rate, $\dot{\phi }$ is the roll rate, b is the track width, $h_{cg}$ is the centre of gravity height $h_{r}$ is the roll centre height, $h_{p}$ is the pitch centre height, and $l_{f}$ and $l_{r}$ are the distance from the front and rear axle to the vehicle CoG, respectively (see Figure 4). Bechtloff [208] has used experimental tests on a banked corner of 30 degrees to show that the gyroscopic term can be neglected. Road slope and banked corners may be taken into account for normal force calculation [202]:(34)$$F_{z,fl}=\frac{l_{r}}{2L}mgcos\left(\eta_{long}\right)cos\left(\eta_{lat}\right)-\frac{1}{2L}mgh_{cg}sin\left(\eta_{long}\right)-\frac{l_{r}}{BL}mgh_{cg}sin\left(\eta_{lat}\right)\;-\frac{1}{2L}h_{cg}ma_{x}-\frac{l_{r}}{BL}h_{cg}ma_{y}-\frac{1}{2L}\;\rho C_{x}A_{x}V_{x}^{2}h_{d},$$
(35)$$F_{z,fr}=\frac{l_{r}}{2L}mgcos\left(\eta_{long}\right)cos\left(\eta_{lat}\right)-\frac{1}{2L}mgh_{cg}sin\left(\eta_{long}\right)+\frac{l_{r}}{BL}mgh_{cg}sin\left(\eta_{lat}\right)\;-\frac{1}{2L}h_{cg}ma_{x}+\frac{l_{r}}{BL}h_{cg}ma_{y}-\frac{1}{2L}\;\rho C_{x}A_{x}V_{x}^{2}h_{d},$$
(36)$$F_{z,rl}=\frac{l_{f}}{2L}mgcos\left(\eta_{long}\right)cos\left(\eta_{lat}\right)+\frac{1}{2L}mgh_{cg}sin\left(\eta_{long}\right)-\frac{l_{f}}{BL}mgh_{cg}sin\left(\eta_{lat}\right)\;+\frac{1}{2L}h_{cg}ma_{x}-\frac{l_{f}}{BL}h_{cg}ma_{y}+\frac{1}{2L}\;\rho C_{x}A_{x}V_{x}^{2}h_{d},$$
(37)$$F_{z,rr}=\frac{l_{f}}{2L}mgcos\left(\eta_{long}\right)cos\left(\eta_{lat}\right)+\frac{1}{2L}mgh_{cg}sin\left(\eta_{long}\right)+\frac{l_{f}}{BL}mgh_{cg}sin\left(\eta_{lat}\right)+\;\frac{1}{2L}h_{cg}ma_{x}+\frac{l_{f}}{BL}h_{cg}ma_{y}+\frac{1}{2L}\;\rho C_{x}A_{x}V_{x}^{2}h_{d},$$
where $\eta_{long/lat}$ are longitudinal and lateral road slopes. Forces presented in Equations (28)–(33), and (34)–(37) are presented in Figure 5.

The most widespread controller type for lateral vehicle dynamics includes longitudinal or lateral forces and yaw moment. It is used for ESC systems performing using braking forces and can be calculated as follows:(38)$$M_{ESC}=\left(F_{x,fr}-F_{x,fl}\right)\frac{b}{2}+\left(F_{x,rr}-F_{x,rl}\right)\frac{b}{2}.$$

This formula can be used for ESC systems based on TV where moments are generated from electric motors as well. Systems where the ESC function is realised using steering can be defined as follows:(39)$$M_{ESC}=F_{y,f}l_{f}-F_{y,r}l_{r}.$$

Feedback control is widely used to control the vehicle. It is a control system that continually monitors a system’s output and makes adjustments to maintain it close to a desired reference, ensuring stability and performance. To realise a feedback control strategy, the error between the reference and measured/estimated values (presented in Section 4.1 and Section 4.2) is achieved and the demand is calculated. The control error can be defined as follows [201]:(40)$$e_{x}=\left\{\begin{array}{c} x_{ref}-x_{meas/est},\left|x_{ref}-x_{meas/est}\right|>\Delta_{x}\\ 0,\left|x_{ref}-x_{meas/est}\right|<\Delta_{x} \end{array}\right.,$$
where $x_{ref}$ is the reference value and $x_{meas/est}$ is the measured or estimated value. Thresholds $\Delta_{x}$ are used to define dead zones to eliminate the demand generation when control errors are insignificant, for each parameter it is defined separately.

The control law for ESC for PID controller can be described as:(41)$$M_{ESC}=\Delta M_{\psi }=K_{p}e_{\dot{\psi }}+K_{i}\int e_{\dot{\psi }}dt+K_{d}\Delta e_{\dot{\psi }},$$
where $K_{p}$ is proportional, $K_{i}$ is integral, $K_{d}$ is differential gains, $e_{\dot{\psi }}$ is the yaw rate error calculated using Formula (40).

Similarly, control law can be described using sideslip angle as a reference:(42)$$M_{ESC}=\Delta M_{\psi }=K_{p}e_{\beta }+K_{i}\int e_{\beta }dt+K_{d}\Delta e_{\beta },$$
where $e_{\beta }$ is the sideslip angle error, which is defined in a more complex way compared to (40) [201]:(43)$$e_{\beta }=\left\{\begin{array}{c} \left|\beta_{est}-\beta_{ref}\right|sign\left(\beta_{est}\right),\beta_{est}>\beta_{ref}\wedge {\dot{\beta }}_{ref}>0\;\\ \begin{array}{c} \left|\beta_{est}-\beta_{ref}\right|sign\left(\beta_{est}\right),\beta_{est}<\beta_{ref}\wedge {\dot{\beta }}_{ref}<0\\ 0,\;otherwise \end{array} \end{array}.\right.$$

The above-presented approaches can be used for ESC systems based on braking, TV, and brake blending. In systems where the ESC function is realised using steering, the required moment can be calculated as follows:(44)$$M_{ESC}=\Delta M_{\psi }=F_{y,f}l_{f}-F_{y,r}l_{r}.$$

In some cases, instead of forces or moments other parameters may be defined. For example, in AV with four-wheel steering, demand for realisation ESC is calculated as follows:(45)$$\Delta \delta =\Delta \delta_{f}-\Delta \delta_{r}=\left(\frac{l}{V}+K_{us}V_{x}\right)e_{\dot{\psi }},$$
where $e_{\dot{\psi }}$ is the yaw rate error and $K_{us}$ is the understeer gradient. Four-wheel steering can be realised using four toe actuators. Such architecture can reduce Ackermann steering error, or even switch to the Anti-Ackermann steering while driving on handling limit.

The control law for front wheels can be represented as follows:(46)$$\Delta \delta_{f}=K_{p}e_{\dot{\psi }}+K_{i}\int e_{\dot{\psi }}dt+K_{d}\Delta e_{\dot{\psi }}.$$

The positioning of rear wheels can be done by proportional control, as shown in [41]:(47)$$\Delta \delta_{r}=k\left(V\right)\Delta \delta_{f},$$
where $k\left(V\right)$ is steering gain, as presented in Figure 6.

The control strategies presented above for vehicle lateral dynamics control. Longitudinal vehicle dynamics controllers used for ABS or traction control systems (TCS) are based on vehicle wheel slip, using slip error $e_{\lambda }$ as input:(48)$$M_{br/tr}=K_{p}e_{\lambda }+K_{i}\int e_{\lambda }dt+K_{d}\Delta e_{\lambda }.$$

Slip error is defined using (40). In this example, PID controller is shown.

AD requires additional sensors like cameras, radars, and lidars to realise perception [209], as mentioned in Section 3. AD provides great opportunities for the improvement of vehicle control by implementing feedforward control as a preview option granted by the perception system. First, feedforward control can be realised for steering to provide an estimate of the steering angle required to traverse a path with a known path curvature and velocity [210]. This minimises the level of compensation required by the steering feedback, reducing tracking errors and allowing for less overall control effort [210]. Below, the example is provided for the front wheels steering vehicle.

The required steering angle can be defined as follows [211]:(49)$$\delta =\delta_{fb}+\delta_{ff}=K_{py}e_{y}+K_{p\psi }e_{\psi }+\delta_{ff},$$
where $\delta_{fb}$ is the steering angle required by feedback control, $\delta_{ff}$ is the steering angle required by feedforward control, $e_{y}$ is lateral error, $e_{\psi }$ is heading error, and $K_{py}$ and $K_{p\psi }$ are proportional gains.

Feedforward contribution is calculated based on the turn radius R:(50)$$\delta_{ff}=\frac{\left(L+K_{us}V_{x}^{2}\right)}{R}.$$

Preview system parameters $e_{y},e_{\psi },R$ can be defined using computer vision. When the vehicle in the front blocks the line of sight of the camera, only the longitudinal and lateral distance to the preceding vehicle can be accurately obtained, and single point information is available instead of path information. For this reason, two feedback control laws are applied. Switching between these laws is done depending on the available measurements.

In the case of PT, all three system parameters ($e_{y},e_{\psi },R$) are available. Combined feedback error can be defined using $x_{LA}$ and $d_{LA}$ (Figure 4), and the steering angle can be defined as follows [211]:(51)$$\delta =\left(L+K_{us}V_{x}^{2}\right)\left(\frac{2}{d_{LA}^{2}}e_{y,road}+\frac{2}{d_{LA}^{2}}\left(e_{y}+x_{LA}e_{\psi }\right)\right).$$

Road curvature at the vehicle’s centre of gravity can be approximated from:(52)$$\hat{R}=\frac{d_{LA}^{2}}{2e_{y,road}}.$$

The aforementioned examples are dependent on positioning infrastructure like lane markings and GNSS for PT. However, recent advancements in PT systems are geared toward addressing more complex scenarios, including both unsignalised and signalised intersections [212] and multi-lane roads. These advanced systems also require robust perception capabilities to detect road signs and traffic lights. In addition, these PT systems must operate effectively in situations where traditional positioning infrastructure is unavailable. In such circumstances, the system must: (i) localise multiple vehicles within a shared coordinate system, (ii) maintain the desired platoon configuration by reconstructing the historical trajectory of the leading vehicle—this historical trajectory serves as a basis for planning the target state for the following vehicle, and (iii) implement a virtual controller-based algorithm capable of generating feasible trajectories for the following vehicle in real-time [213]. These developments mark a significant shift toward more versatile and robust PT solutions that can handle a wide range of real-world scenarios, making them particularly valuable in novel PT tasks.

Additionally, during AD, there is no driver, only an occupant, who may not be ready for the manoeuvre. So, all the manoeuvres need to be performed ensuring maximum comfort for passengers. To ensure this with the preview system, road slopes and banked corners may be considered. For vertical dynamics, control pitch/roll moments, or normal forces are used [206]. The required force demand can be calculated using (30) and (34)–(37).

Feedforward control can be implemented without a preview option as well. The first case is when feedback control is too slow. Second, measurement/estimation of values needed for error calculation is a complex task, and the “correction” effect that feedback control will provide will not be significant enough. Below we present examples of such systems.

Using active suspension components, handling and comfort may be improved, not only by changing damping forces, but changing pitch and roll moments thus minimising pitch and roll angles. There are solutions for roll angle estimation, for example using Takagi-Sugeno fuzzy observer [214]. Practically, pitch and roll angle measurement is problematic on production vehicles, where the IMU sensor is used, which provides only angle rates. So, it may be challenging to realise feedback control. As a result, feedforward algorithms can be used for such tasks. The active anti-roll moment can be realised as a linear function of the lateral acceleration [66]:(53)$$M_{\phi }=m_{spr}a_{y}\left(h_{cg}-h_{r}\right),$$
where $m_{spr}$ is vehicle SM.

The active anti-pitch moment is obtained from a linear function of the longitudinal acceleration:(54)$$M_{\theta }=m_{spr}a_{x}\left(h_{cg}-h_{p}\right).$$

Realising control strategies acceleration is used as an input parameter for pitch and roll control. 

The control law for pitch and roll can be defined as:(55)$$M_{pitch/roll}=K_{p}a_{x/y,ref}+K_{i}\int a_{x/y,ref}dt+K_{d}\Delta a_{x/y,ref},$$
where $a_{x/y,ref}$ refers to the reference accelerations in longitudinal and lateral directions.

Practically, the values of pitch and roll angles will not be equal to zero without feedback control, as deformation of the tyres appears; however, the error in practical systems is less than 1 degree. If such accuracy is not enough, pitch and roll angle can be calculated using data from displacement sensors installed on Ums. They are typically used on production vehicles equipped with semi-active suspensions.

As shown, for ICC in AD feedforward, feedback and a combination of feedforward-feedback controllers may be used. The comparison of various control methods is shown in Table 2. All of them have advantages and disadvantages, and researchers developed the majority of modifications for controller improvement including, SMC, MPC, optimal, and others [215]. MPC controller modifications which are used for PT include adaptive MPC, linear time-varying MPC, nonlinear MPC, hybrid MPC, ANN MPC, robust MPC, and learning MPC; more details can be found in [101,216,217]. More information regarding control methods can be found in [102,103,218]. The usage of adaptive MPC together with stability systems such as direct yaw moment control increases PT accuracy [216].

The system control variables are determined by the high-level controller, denoted as $v={\left[F_{x},\Delta F_{z},\delta ,\Delta M_{\psi }\right]}^{T}$ in this context. Specifically, the longitudinal force $F_{x}$ is the primary variable essential for longitudinal dynamics, and it can be substituted with the moment described in Equation (48). The change in vertical force $\Delta F_{z}$ can be computed using Equations (34)–(37), or it can be replaced with the pitch/roll moments introduced in Equations (53) and (54). The use of pitch and roll moments necessitates additional normal forces to enhance ride comfort. The steering angle δ is employed for PT integration, as outlined in Formula (51), and it can be adjusted with Δδ (Equation (45)) to enhance lateral dynamics. This can be implemented for both front- and four-wheel driving scenarios. Alternatively, the steering angle can be substituted with the moment described in Equation (39); however, this is a more intricate solution. Furthermore, required yaw moments can be generated as described in Equations (41) and (42).

### 4.4. Middle-Level Control

Modern ground vehicles may be over-actuated systems consisting of several active sub-systems such as brakes, steering, suspension, individual-wheel electric motors, wheel positioning (camber, toe), and active aerodynamics actuators. Each of the actuators is independently developed to achieve a specific goal. Improper integration of several subsystems leads to overlapping regions in control tasks. Therefore, to handle such an over-actuation and to prevent control objectives interference between subsystems, an algorithm is required to allocate the control actions of the different actuators [2,238].

Formulating CA involves computationally intensive tasks and can be categorised into various types: non-optimal methods, linear programming (LP), quadratic programming (QP), nonlinear programming (NLP), model predictive CA (MPCA), multi-agent system (MAS) based, data-driven, and hybrid. Numerous methods have been developed for each of these CA types; more details can be found in [107,239,240,241].

The optimisation-based weighted least-squares CA method, falling under the QP category, is predominantly utilised in ICC implementations [241]. The CA problem for ICC can be formulated as an optimisation task to minimise control input and allocation error [38]:(56)$$u=\arg\underset{\underline{u}\leq u\leq \bar{u}}{\underset{⏟}{\min}}\left(\|W_{u}{\left(u-u_{des}\right)\|}_{2}^{2}+\gamma \|W_{v}{\left(Bu-v\right)\|}_{2}^{2}\right)\;=\arg\underset{\underline{u}\leq u\leq \bar{u}}{\underset{⏟}{\min}}\left(\|W_{u}\Delta {u\|}_{2}^{2}+\gamma \|W_{v}{\left(Bu-v\right)\|}_{2}^{2}\right),$$
where v is the virtual control input request from the high-level controller, which includes [$F_{x},\Delta F_{z},\delta ,\Delta M_{z}]$, u is the actual (reference) control input, with $\underline{u}\leq u\leq \bar{u}$ actuation limits, B is the control effectiveness matrix, $W_{u}$ and $W_{v}$ are the weighting matrices for penalising the use of specific actuators and for penalising the specific virtual control input, γ is the weighting parameter to minimise the allocation error, and $u_{des}$ is the desired control input.

The objective function from (56) can be modified by adding additional objectives in the cost function as follows:(57)$$J\left(k\right)=\overset{p}{\underset{i=1}{\sum }}\|u\left(k+i\right)-v{\left(k+i\right)\|}_{2}^{2}+\overset{p}{\underset{i=1}{\sum }}\|\Delta u{\left(k+i\right)\|}_{2}^{2}+J_{add},$$
where $J_{add}$ refers to additional objectives. A description of this part is presented in Table 3. Some of the additional objectives may be used in combinations, and such an approach requires weights for each term. There are additional objectives, where the mechanical loss of actuators can be taken into account as well [242]. Partial cost functions that may be used for ICC realisation in AVs are presented in Table 4.

Choosing the appropriate weights for the cost function, as outlined in Formula (56), is a crucial step in enabling the pursuit of diverse strategies such as comfort, stability, and safety, either individually or in a combined manner, while concurrently minimising overall energy consumption. Prioritising safety is paramount, and this can be achieved through various means such as emergency braking or lane change manoeuvres. Typically, the cost function is structured to ensure that the sum of weights equals 1. When making decisions regarding the prioritisation of safety, stability, and comfort, the weight assigned to safety should be approximately one order of magnitude higher than the weight assigned to the next priority, such as stability [250]. 

Several procedures exist for identifying weights, with the first being based on predefined key performance indicators (KPIs). Various combinations of weights are proposed, and simulations are conducted to compare the achieved KPIs, allowing for the selection of appropriate weights. Another approach involves weight identification techniques based on fuzzy logic [251], or the implementation of operational research methods for weight definition [252]. Additionally, weights may be presented as hyperbolic tangent or exponential functions [253].

During the realisation of the control strategy, it is essential to take into account limitations, which are caused by actuators, vehicle physical limits (saturation), or comfort requirements. Saturation and comfort requirements for longitudinal velocity, longitudinal and lateral accelerations, yaw rate, and slip angle were presented in Section 4.2. Here we focus on the physical limits of the actuators and requirements for PT. For the steering system, the steering angle is [101]:(58)$$\delta_{min}\leq \delta \leq \delta_{max},$$
and steering rate is:(59)$${\dot{\delta }}_{min}\leq \dot{\delta }\leq {\dot{\delta }}_{max}.$$

Similarly, there are limited camber and toe actuators, braking and propulsion torques and their rates, and normal forces produced by active suspension components and/or aerodynamic actuators. For PT errors are limited as follows [101]: (60)$$e_{y,min}\leq e_{y}\leq e_{y,max},$$
and
(61)$$e_{\psi ,min}\leq e_{\psi }\leq e_{\psi ,max}.$$

Distance to the obstacle is limited as follows:(62)$$X_{min}\leq X\leq X_{max}.$$

In recent years, Model Predictive Control Allocation (MPCA), an optimal control technique based on receding horizon control, has gained popularity among researchers, especially in addressing PT tasks [101]. This approach allows us to consider the predicted future behaviour of the system in the optimisation problem. The cost function is formulated as follows [1]: (63)$$\underset{U}{\min}J\left(X\left(0\right),U\left(\cdot \right)\right)=l_{Np}\left(X\left(N_{p}\right)\right)+\overset{N_{c}-1}{\underset{k=0}{\sum }}l\left(X\left(k\right),U\left(k\right)\right)+\overset{N_{p}-1}{\underset{k=N_{c}}{\sum }}l\left(X\left(k\right),U\left(N_{c}-1\right)\right),$$
where $X\left(0\right)$ is the initial value of state vector, $U\left(\cdot \right)$ is the control sequence, $l_{Np}\left(X\left(N_{p}\right)\right)$ is the terminal cost, $\overset{N_{c}-1}{\underset{k=0}{\sum }}l\left(X\left(k\right),U\left(k\right)\right)$ is the stage cost function associated with each time step, $N_{c}$ is the number of steps in the control horizon, and $N_{p}$ is the number of steps in the prediction horizon. More details can be found in [1]. 

Previously described CA methods rely on the system model [254]. Alternatively, an ANN-based approach or data-driven approach has been proposed for CA tasks [254,255]. This approach allows for the training of the model using input-output data without necessitating knowledge of the system dynamics. Furthermore, such model-based approaches are applicable in real-time applications, presenting a potential advantage over methods like MPCA, which may pose challenges in certain real-time scenarios.

The final layer of the coordinated control structure is low-level controllers. Low-level control is used to generate input signals to control the actuator and to calculate constraints for middle-level CA. In production vehicles, the price of computational hardware is critically important; at the same time, functional safety requirements must be warranted. As a result, to realise low-level control, three main approaches are used widely: lookup tables, rule-based algorithms, and variations of PID algorithms. Commonly, these algorithms are designed and validated by Tier 1 and Tier 2, who design actuators and/or ECUs, and together, they ensure the functional safety of the system and perform homologation procedures.

## 5. Discussion and Conclusions

In the domain of ICC, the utilisation of X-in-the-loop environments or vehicle demonstrators remains limited, as evidenced by a constrained number of research papers [13,256,257,258,259,260,261]. The prevailing trend in ICC investigations predominantly involves simulation-based methodologies. While sophisticated control strategies have been conceptualised, uncertainties persist regarding their real-time applicability. Computational power has traditionally limited the seamless integration of advanced ICC algorithms into production vehicles. Noteworthy industry players including OEMs and Tier 1, are orchestrating a paradigm shift in contemporary vehicle architecture. This transition involves moving from multiple individual control units to a streamlined centralised configuration using a few high-performance computers [262,263]. This evolution is already manifested in production vehicles like the Volkswagen ID series [263], heralding promising prospects for ICC advancement.

X-by-wire technology, particularly brake-by-wire, has become indispensable in numerous EVs during the last decade. Steer-by-wire, introduced nearly a decade ago in the Nissan Infinity Q50, is gaining widespread adoption by various OEMs and Tier 1 [264]. Mandatory ABS and ESC systems in new production vehicles, coupled with the availability of actuated suspension systems, collectively contribute to the continuous evolution of ICC.

A comprehensive examination of existing literature underscores a notable division in the analysis between PT tasks and ICC. The amalgamation of these tasks holds substantial potential for various advantages. This includes augmenting the effectiveness of the ESC system through active steering and enhancing PT by bolstering stability with ICC interventions. Prior investigations have predominantly delved into systems concentrating on longitudinal and lateral vehicle dynamics control. However, considerations such as motion sickness and ride comfort have risen as AD gains prominence. Research indicates a concerted effort to attain a comfort level in AV comparable to that experienced in trains [177,265]. A plausible hypothesis emerges, suggesting an anticipated surge in interest in ICC systems, specifically focusing on vertical dynamics in the future.

Within the PT tasks, acquiring initial data for driving trajectories typically occurs without considering perception integration. It is imperative to address this gap. It necessitates a meticulous consideration of the sensors responsible for generating this initial data, a thorough analysis of their accuracy and redundancy, and formulation of algorithms to ensure system resilience in fail-safe mode.

## Figures and Tables

**Figure 1 sensors-24-00600-f001:**
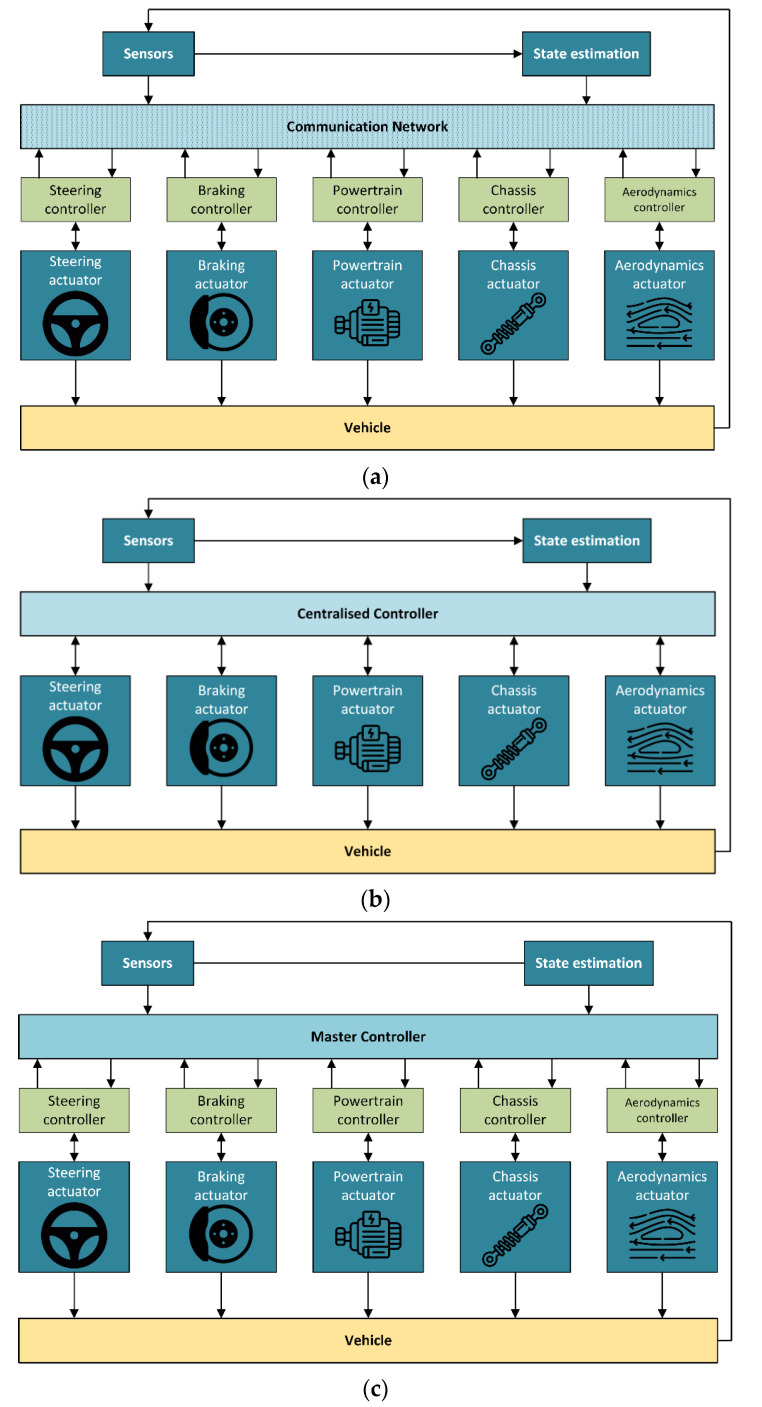
Simplified integrated chassis control structures: (**a**) decentralised, (**b**) centralised, and (**c**) coordinated.

**Figure 2 sensors-24-00600-f002:**
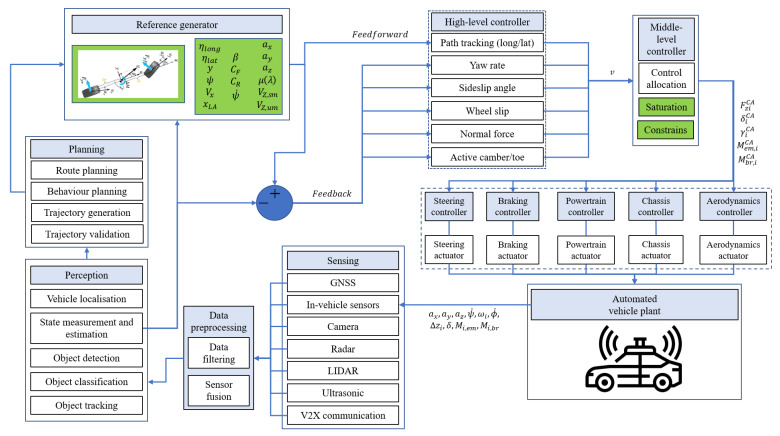
The overall structure of the automated vehicle with integrated chassis control.

**Figure 3 sensors-24-00600-f003:**
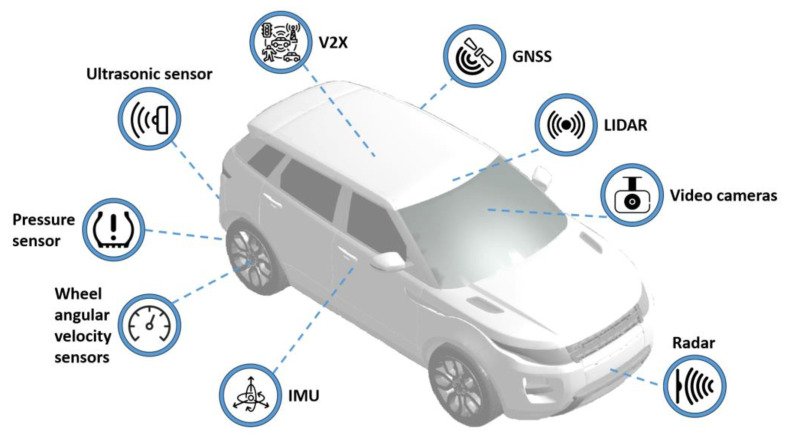
Sensor systems used in ICC and AD.

**Figure 4 sensors-24-00600-f004:**
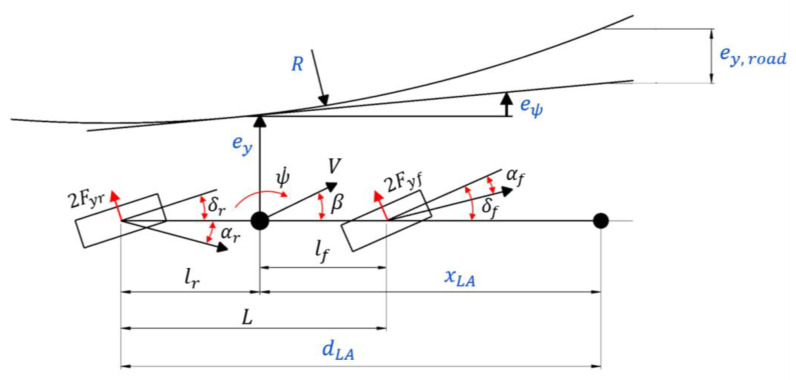
Bicycle model.

**Figure 5 sensors-24-00600-f005:**
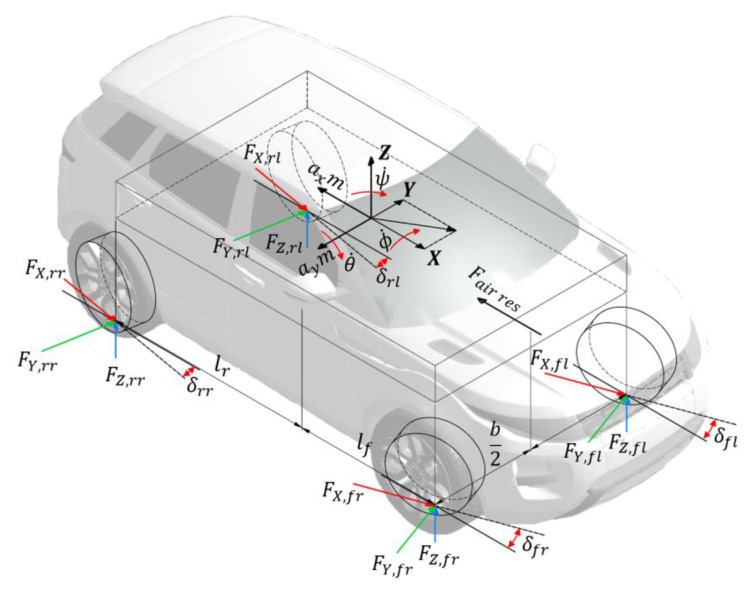
Main forces acting on the vehicle.

**Figure 6 sensors-24-00600-f006:**
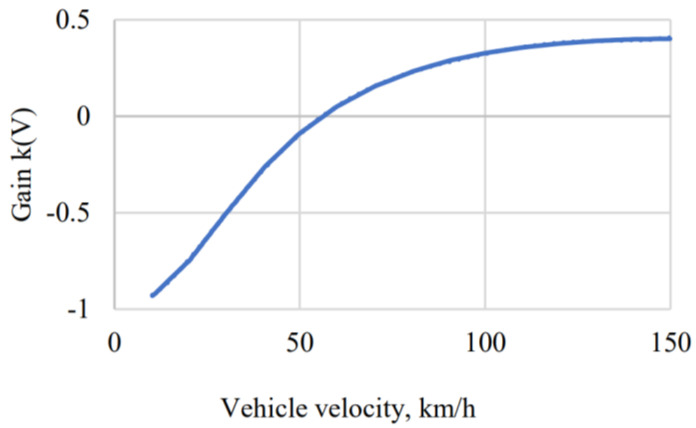
Steering gain of rear wheels (developed by authors, based on [42]).

**Table 1 sensors-24-00600-t001:** Sensor classes used for AD with ICC.

Sensor Classes (Data Source)	Advantages	Disadvantages	Functions
GNSS	global positioning, heading information, time synchronisation, insensitive to weather conditions, and does not require pre-processing	vulnerable to weak and multipath signals, no heading information while stationary, and low accuracy for usual GNSS	global positioning, velocity, and heading
IMU	low cost and independent of weather conditions	requires pre-processing, filtering, and data fusion	vehicle accelerations and relative position, angular velocities of sprung mass (SM), and heading
LIDAR	3D data about environment, objects, high stability, high accuracy, and independent of lighting conditions	high cost, limited mounting point selection (for visibility), complex processing algorithms, high computational resources, dependent on weather conditions (degraded performance under rain, snow and fog), does not work with reflective and transparent (glass) obstacles, and limited longevity due to moving parts and being placed outside the vehicle	object detection, and distance to the object measurement
Radar	independent of weather conditions, easily extractable information about the distance and velocity of objects, and can be placed behind plastic parts	does not detect some objects (that do not reflect millimetre-length waves or microwaves), complex processing algorithms, and provides noisy output	distance measurement, velocity movement direction, and azimuth and elevation of objects
Visual sensors (monocular or stereo cameras)	low cost, long life, can be placed inside the vehicle, high resolution, allows environmental and situational awareness, and abundance of information	requires complex data processing algorithms, high computational power requirements, depends on lightning and visibility conditions, and limited sampling frequency motion blur	object detection, object classification, object tracking, visual odometry, visual localisation, road curvature and geometry, and weather conditions
Thermal cameras	independent of lighting conditions	high price, low resolution, complex processing algorithms, limited sampling frequency, high computational power requirements, and data dependency on thermal conditions	object detection, object classification, and object tracking
Ultrasonic sensors	low cost, fast processing, low power consumption, insensitive to weather conditions, and can be placed behind plastic parts	short working range and sensitive to sensor contamination, e.g., dust or snow	local proximity
Other in-vehicle sensors	direct measurement	limited sampling frequency	wheel encoders, braking pressure, steering angle, UM vertical displacement/acceleration, tire pressure, temperature
Estimators and virtual sensors (VSs)	upgradable, no recurring production costs, and reduced mechanical complexity of the vehicle	requires a mathematical model or dataset, accuracy depends on the accuracy of the model or coverage of data, high development cost, and requires computational resources	velocity, sideslip angle, tire pressure, traction torque, and SM/UM velocity
V2X communication	low power consumption, low computational resources, and enables information sharing	cybersecurity issues and wide variety of standards and protocols	driving restrictions, road condition, preliminary location, traffic information, ambient conditions, and states of the vehicles and other traffic participants

**Table 2 sensors-24-00600-t002:** Advantages and disadvantages of different control methods.

Control Method	Advantages	Disadvantages	Source
Rule-based	Used for their high functional safety, low computational cost, and real-time capability	Commonly outperformed by other controllers and the complexity increases significantly for more complex tasks.	[219,220]
PID	Simple structure, easy implementation, and robustness	Difficult to tune, contradiction between overshooting and response time, and poor versatility	[103,104,218,221,222]
Backstepping (BS) Nonlinear BS	Robustness can be used for complex and nonlinear systems control	Requires the design of Lyapunov functions and has high computational complexity	[223]
Pure pursuit (Stanley) (only for PT)	Simple layout, few predictable parameters, suitability for controlling vehicle position	Poor adaptability to nonlinear systems (thus not suitable for high velocities) and road curvatures because adjusting the look ahead distance is challenging.	[103,218]
SMC	Few adjustment parameters, fast response, insensitivity to disturbances, and parameter change	It has a problem with the chattering phenomenon	[103,218,224,225,226,227]
$H_{\infty }$	Easy to establish constraints and has strong robustness	Requires complicated theoretical derivation and can handle only bounded disturbances	[103,228,229,230]
MPC	A controller can properly deal with multiple state and actuator constraints	Difficult to analyse system stability, high computational complexity, and poor real-time performance	[1,218,231,232]
ANN	Good adaptability for nonlinear systems and can be used in parallel with other controllers to reduce computational requirements	Needs training datasets for each new task and needs to be retrained for each new vehicle	[218,233,234,235]
Fuzzy	Few requirements for mathematical model accuracy	The selection of rules is not systematic and it is difficult to correct tracking errors quickly	[104,218,236]
Optimal	Performance indicators may be optimised	High requirements for mathematical model accuracy and is thus complex to implement for control of nonlinear systems. It also has poor robustness.	[103,177,218,237]

**Table 3 sensors-24-00600-t003:** Additional objectives in cost functions.

Additional Objective	Formulation	Source
Tyre energy dissipation longitudinal	$J_{add}=\sum_{i=1}^{4}\int_{0}^{T}F_{x,i}V_{\lambda ,i}dt$ $=\sum_{i=1}^{4}F_{x,i}^{2}V_{y,i}^{2}$	[97]
Tyre energy dissipation	$J_{add}=\overset{4}{\underset{i=1}{\sum }}F_{x,i}^{2}V_{x,i}^{2}+\overset{4}{\underset{i=1}{\sum }}F_{y,i}^{2}V_{y,i}^{2}$	[243]
Wheel slip power losses	$J_{add}=\overset{4}{\underset{i=1}{\sum }}\int_{0}^{T}P_{w,i}\left(1-\lambda_{i}\right)dt$	[97]
Energy consumption due to slip	$J_{add}=\overset{4}{\underset{i=1}{\sum }}\left|F_{x,i}\right|$	[97]
Tyre wear	$J_{add}=\overset{4}{\underset{i=1}{\sum }}\left|\alpha_{i}\right|$	[97]
Friction rate or tyre workload	$J_{add}=\frac{\sqrt{F_{x,i}^{2}+F_{y,i}^{2}}}{F_{\mu ,i}}$ $\mathrm{Defining}\;F_{\mu ,i}$ may be challenging so different objectives may be reformulated as: $J_{add}=\overset{4}{\underset{i=1}{\sum }}\mu_{i}^{2}=\overset{4}{\underset{i=1}{\sum }}\frac{F_{x,i}^{2}+F_{y,i}^{2}}{F_{z,i}^{2}}$	[43,243]

where T is the manoeuvre duration, $V_{x,i}$ is the wheel linear velocity, i=fl,fr,rl,rr; $\alpha_{i}$ tyre slip angle, $P_{w,i}$ is the power supplied to the wheel I, $\lambda_{i}$ is a slip of wheel I, $F_{x,\;i}$ is longitudinal, and $F_{y,\;i}$ is lateral tyre forces, and $F_{\mu ,i}$ is the force defined from the friction circle radius.

**Table 4 sensors-24-00600-t004:** Partial cost function.

Path Tracking		Source
Longitudinal	$\mathrm{Velocity}\;\mathrm{offset}:\;J=\int_{t_{s}}^{t_{f}}{\left(V_{x,ref}-V_{x}\right)}^{2}dt$ $\mathrm{Longitudinal}\;\mathrm{acceleration}:\;J=\int_{t_{s}}^{t_{f}}a_{x}^{2}dt$ $\mathrm{Longitudinal}\;\mathrm{acceleration}\;\mathrm{error}:\;J=\int_{t_{s}}^{t_{f}}\left(a_{x,ref}-a_{x}\right)dt$ $\mathrm{Jerk}:\;J=\int_{t_{s}}^{t_{f}}{\dot{a}}_{x}^{2}dt$ $\mathrm{Lateral}\;\mathrm{position}\;\mathrm{error}:\;J=\int_{t_{s}}^{t_{f}}{\left(x_{ref}-x\right)}^{2}dt$ $\mathrm{Path}\;\mathrm{length}:\;J=\int_{t_{s}}^{t_{f}}V_{x}dt$	[101,244,245]
Lateral	$\mathrm{Lateral}\;\mathrm{position}\;\mathrm{error}:\;J=\int_{t_{s}}^{t_{f}}{\left(y_{ref}-y\right)}^{2}dt$ $\mathrm{Heading}\;\mathrm{angle}\;\mathrm{error}:\;J=\int_{t_{s}}^{t_{f}}{\left(\psi_{ref}-\psi \right)}^{2}dt$ $\mathrm{Steering}\;\mathrm{angle}:\;J=\int_{t_{s}}^{t_{f}}\delta^{2}dt$ Steering rate: $J=\int_{t_{s}}^{t_{f}}{\dot{\delta }}^{2}dt$	[101,244,245]
Terminal Cost		
ICC		
Longitudinal	$\mathrm{Energy}:\;J=\int_{t_{s}}^{t_{f}}P_{x}^{2}dt$ $\mathrm{Wheel}\;\mathrm{slip}:\;J=\int_{t_{s}}^{t_{f}}{\left(\lambda_{ref}-\lambda \right)}^{2}dt$ $\mathrm{Wheel}\;\mathrm{slip}\;\mathrm{power}\;\mathrm{loss}:\;J=\int_{t_{s}}^{t_{f}}{\left(P_{ref}-M_{i}\omega_{i}\lambda_{i}\right)}^{2}dt$ $\mathrm{Traction}\;\mathrm{moment}:\;J=\int_{t_{s}}^{t_{f}}M_{tr}^{2}dt$ $\mathrm{Braking}\;\mathrm{moment}:\;J=\int_{t_{s}}^{t_{f}}M_{br}^{2}dt$	[22,97,246]
Lateral	$\mathrm{Yaw}\;\mathrm{rate}\;\mathrm{error}:\;J=\int_{t_{s}}^{t_{f}}{\left({\dot{\psi }}_{ref}-\dot{\psi }\right)}^{2}dt$ $\mathrm{Sideslip}\;\mathrm{error}:\;J=\int_{t_{s}}^{t_{f}}{\left(\beta_{ref}-\beta \right)}^{2}dt$ Lateral acceleration: $J=\int_{t_{s}}^{t_{f}}a_{y}^{2}dt$ $\mathrm{Lateral}\;\mathrm{acceleration}\;\mathrm{error}:\;J=\int_{t_{s}}^{t_{f}}\left(a_{y,ref}-a_{y}\right)dt$ Yaw moment: $J=\int_{t_{s}}^{t_{f}}M_{\psi }^{2}dt$	[97,244,246]
Vertical	$\mathrm{Sprung}\;\mathrm{mass}\;\mathrm{acceleration}:\;J=\int_{t_{s}}^{t_{f}}{\ddot{z}}_{SM}^{2}dt$ $\mathrm{Pitch}\;\mathrm{acceleration}:\;J=\int_{t_{s}}^{t_{f}}{\ddot{\theta }}^{2}dt$ $\mathrm{Roll}\;\mathrm{acceleration}:\;J=\int_{t_{s}}^{t_{f}}{\ddot{\phi }}^{2}dt$ $\mathrm{Yaw}\;\mathrm{acceleration}:\;J=\int_{t_{s}}^{t_{f}}{\ddot{\psi }}^{2}dt$ $\mathrm{Velocity}\;\mathrm{change}\;\mathrm{between}\;\mathrm{SM}\;\mathrm{and}\;\mathrm{UM}:\;J=\int_{t_{s}}^{t_{f}}\left({\dot{z}}_{SM}-{\dot{z}}_{UM}\right)dt$ $\mathrm{Displacement}\;\mathrm{change}\;\mathrm{between}\;\mathrm{SM}\;\mathrm{and}\;\mathrm{UM}:\;J=\int_{t_{s}}^{t_{f}}\left(z_{SM}-z_{UM}\right)dt$ $\mathrm{Displacement}\;\mathrm{change}\;\mathrm{between}\;\mathrm{UM}\;\mathrm{and}\;\mathrm{road}:\;J=\int_{t_{s}}^{t_{f}}\left(z_{um}-z_{road}\right)dt$ $\mathrm{SM}\;\mathrm{vertical}\;\mathrm{velocity}:\;J=\int_{t_{s}}^{t_{f}}{\dot{z}}_{SM}^{2}dt$ $\mathrm{SM}\;\mathrm{vertical}\;\mathrm{displacement}:\;J=\int_{t_{s}}^{t_{f}}z_{SM}^{2}dt$	[177,246,247,248,249]

where $t_{s}$ is the starting time, $t_{f}$ is the finish time, $M_{tr}$ is the traction moment, $M_{br}$ is the braking moment, $M_{\psi }$ is the yaw moment, ${\ddot{z}}_{SM}$ is the SM acceleration, $M_{i}$ is the wheel torque, $\omega_{i}$ is the angular wheel, and $\lambda_{i}$ is the wheel slip.

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
