# Peer review of "Review of Integrated Chassis Control Techniques for Automated Ground Vehicles"

_sensors, 2024, doi:10.3390/s24020600_

Round 1

Reviewer 1 Report

Comments and Suggestions for Authors

This paper provides a comprehensive review of vehicle chassis control. This article has a positive significance for advancing the development of vehicle control. My comments of this paper are as follows:

1) ‘A systematic survey of control techniques and applications in connected and automated vehicles’ is the first review article that considers both automated vehicles and connected automated vehicles simultaneously. Please summarize the contributions of this reference in the introduction, so as to further refine the contributions of this paper.

2) For the vehicle sensing module, detection and object tracking is of great importance. It would be better to include the classical work as follows: hydro-3d: hybrid object detection and tracking for cooperative perception using 3d lidar.

3) MPC is widely used for the intelligent electrical vehicle control, such as: coordinated control of path tracking and yaw stability for distributed drive electric vehicle based on ampc and dyc, planning and tracking control of full drive-by-wire electric vehicles in unstructured scenario. Please highlight the advantages of MPC based on the above work.

4) Please highlight the contribution of this work at the end of the introduction.

Author Response

Dear Editor and Reviewers,

Thank you for your valuable comments. All of them were taken into account, and the changes are highlighted in the paper. You can find the detailed answers below.

Additionally, text, figures and tables were added to improve the clarity and readability of the paper.

Reviewer 1

This paper provides a comprehensive review of vehicle chassis control. This article has a positive significance for advancing the development of vehicle control. My comments of this paper are as follows:

  1. ‘A systematic survey of control techniques and applications in connected and automated vehicles’ is the first review article that considers both automated vehicles and connected automated vehicles simultaneously. Please summarize the contributions of this reference in the introduction, so as to further refine the contributions of this paper.

Answer R1.1: We appreciate the insightful comment and acknowledge the importance of clearly summarizing the contributions of our paper. Previously, it was placed in section 2; right now it was moved to the introduction.

  1. For the vehicle sensing module, detection and object tracking is of great importance. It would be better to include the classical work as follows: hydro-3d: hybrid object detection and tracking for cooperative perception using 3d lidar.

Answer R1.2: Thank you for highlighting the significance of vehicle sensing, detection, and object tracking. We agree with your suggestion and incorporated references to enhance the completeness of our discussion on the vehicle sensing module.

  1. MPC is widely used for the intelligent electrical vehicle control, such as: coordinated control of path tracking and yaw stability for distributed drive electric vehicle based on AMPC and DYC, planning and tracking control of full drive-by-wire electric vehicles in unstructured scenario. Please highlight the advantages of MPC based on the above work.

Answer R1.3: We appreciate your comment regarding adaptive MPC and intelligent electrical vehicle control applications. There is a review paper where variation of MPC for path tracking tasks is investigated, and we cite it in the work. However, using AMPC for path tracking together with direct yaw moment control is exciting; therefore, we added this information to our paper.

  1. Please highlight the contribution of this work at the end of the introduction.

Answer R1.4: Thank you for your suggestion. We addressed this in a revised paper.

Reviewer 2 Report

Comments and Suggestions for Authors

The author has conducted an extensive review of autonomous driving chassis control, and after reading the Abstract and Introduction, I am quite enthusiastic. However, I have some concerns as follows:

1.     In Line 270, why integrating new actuators into such a system after its development can be challenging? Or what challenges will be encountered?

2.     In Line 275, why is the coordinated architecture one of the most perspectives?

3.     In Fig. 2, the block of “reference generator” includes two figures, a 2-DOF and a 7-DOF vehicle dynamics model. Should the author replace 7-DOF with the kinetic model? The figure of 7-DOF does not correspond to the parameters next to it.

4.     The author dedicates a significant portion of the manuscript to the discussion of ADAS sensors such as radar and cameras. However, it remains unclear how this aligns with the paper's title, "Integrated Chassis Control."

5.     The introduced chassis coordination architecture is essential for conflict resolution and control coordination between different tasks, particularly in cases like the coordination of AFS and DYC. However, the author has not provided a comprehensive review of this aspect. The related work can refer to: Liang J, Lu Y, Yin G, et al. A distributed integrated control architecture of AFS and DYC based on MAS for distributed drive electric vehicles[J]. IEEE transactions on vehicular technology, 2021, 70(6): 5565-5577.

6.     In Line 73, I would appreciate further clarification on how ICC addresses redundancy issues, such as redundancy in braking or steering.

7.     The author's emphasis is predominantly on the algorithmic aspects; however, chassis control technology encompasses both software and hardware considerations. I find the description in Figure 2 to be ambiguous and lacking in depth. The related work on software and hardware considerations can refer to: Liang J, Feng J, Fang Z, et al. An energy-oriented torque-vector control framework for distributed drive electric vehicles[J]. IEEE Transactions on Transportation Electrification, 2023.

8.     I posit that chassis control is inherently an engineering task. Nevertheless, the author's review appears to lean excessively towards an academic perspective.

Comments on the Quality of English Language

 Minor editing of English language required

Author Response

Reviewer 2

The author has conducted an extensive review of autonomous driving chassis control, and after reading the Abstract and Introduction, I am quite enthusiastic. However, I have some concerns as follows:

  1. In Line 270, why integrating new actuators into such a system after its development can be challenging? Or what challenges will be encountered?

Answer R2.1: We appreciate your enthusiasm and value your concerns. In response to your first question regarding integrating new actuators into a developed system, we explained the challenges more thoroughly.

There are two main aspects. First, when adding a new actuator, additional computational power is required, which can be a challenge. In the case of a centralised controller the increase of computational power is much lower, as there is a low-level controller for the additional actuator.

The second point is that Tier 1 and Tier 2 are not interested in sharing their algorithms, and OEM often has limited competence and human resources for algorithm development and homologation.

  1. In Line 275, why is the coordinated architecture one of the most perspectives?

Answer R2.2: Regarding the question on the coordinated architecture in Line 275, we elaborated on why a coordinated architecture is considered one of the most promising approaches in the context of autonomous driving chassis control to address your concern. The explanation was added to the text, and is presented below as well.

Using a decentralised control structure, it is impossible to realise ICC, as all the controllers work independently. Centralised control structure, has only one controller, and cannot practically conform to the existing vehicle control system development due to (i) lack of modularity, which requires the OEM to develop the controller together with Tier 1 and Tier 2, (ii) complexity of controller, (iii) lack of flexibility when additional actuators or functionalities are needed, (iv) system failure in a case of controller malfunction [1, doi: 10.1109/TVT.2021.3076105]. This work focuses on coordinated architecture as one of the most perspective ones.

  1. In Fig. 2, the block of “reference generator” includes two figures, a 2-DOF and a 7-DOF vehicle dynamics model. Should the author replace 7-DOF with the kinetic model? The figure of 7-DOF does not correspond to the parameters next to it.

Answer R2.3: Thank you for pointing this out. A simple bicycle model is used for active steering, and a linear / nonlinear bicycle model is needed for path tracking, depending on the handling regime. 7-DOF model is required for TV and brake blending to realise DYC. We reviewed Figure 2 and made corrections based on the reviewer's comments. Additionally, a description for Figure 2 was added.

  1. The author dedicates a significant portion of the manuscript to the discussion of ADAS sensors such as radar and cameras. However, it remains unclear how this aligns with the paper's title, "Integrated Chassis Control."

Answer R2.4: Thank you for pointing this out. We added additional figures and tables to make this section more clear. The idea is that sensors used for automated driving can improve integrated chassis control, and provide a lot of additional data.

For example, having road data from the camera may provide information regarding friction coefficient, which is available in conventional vehicles only after braking. It can be used for active steering systems to realise direct yaw control without braking or torque vectoring. Another system is active suspension with a preview, where the system may be activated before reaching the specified point. As a result, advanced sensors may improve the efficiency of different actuators and integrated chassis control.

  1. The introduced chassis coordination architecture is essential for conflict resolution and control coordination between different tasks, particularly in cases like the coordination of AFS and DYC. However, the author has not provided a comprehensive review of this aspect. The related work can refer to: Liang J, Lu Y, Yin G, et al. A distributed integrated control architecture of AFS and DYC based on MAS for distributed drive electric vehicles[J]. IEEE transactions on vehicular technology, 2021, 70(6): 5565-5577.

Answer R2.5: Thank you for pointing out the need for a more comprehensive review of the introduced chassis coordination architecture. In our revised manuscript, we included a more detailed review, referencing the mentioned paper on the distributed integrated control architecture of AFS and DYC in Sections 3 and 4.

  1. In Line 73, I would appreciate further clarification on how ICC addresses redundancy issues, such as redundancy in braking or steering.

Answer R2.6: We appreciate your request for further clarification on ICC addressing redundancy issues, especially in braking or steering. Below, we explain using direct yaw moment control and active steering system as an example. Direct yaw control can be realised using braking (friction brakes, electric motor, and blending) or powertrain (active differentials or e-motor torque vectoring). Additionally, it can be realised using active steering. The control allocation problem for ICC is formulated as an optimisation task (Section 4.4 of the paper). The cost function contains weights that may be defined differently. Commonly, the prioritisation of safety, stability, and comfort are performed. Additionally, energy consumption can be taken into account. The weight assigned to safety should be approximately one order of magnitude higher than the weight assigned to the next priority, such as comfort. This explanation is provided in Sections 4.3 and 4.4.

Additionally, it can be realised in the same manner proposed in the suggested paper.

  1. The author's emphasis is predominantly on the algorithmic aspects; however, chassis control technology encompasses both software and hardware considerations. I find the description in Figure 2 to be ambiguous and lacking in depth. The related work on software and hardware considerations can refer to: Liang J, Feng J, Fang Z, et al. An energy-oriented torque-vector control framework for distributed drive electric vehicles[J]. IEEE Transactions on Transportation Electrification, 2023.

Answer R2.7: We got acquainted with the proposed paper, found that it is relevant to our investigation, and cited it. Thank you.

Regarding Figure 2, it was modified according to the reviewer's comments. However, such representation of vehicle control structure is typical, an example for conventional vehicles can be found:

https://doi.org/10.1016/j.mechatronics.2014.12.003

doi: 10.1109/ACCESS.2015.2496108

doi: 10.1109/TIE.2016.2540584

The main difference is that, in our case, we modified it, considering the functionality of automated driving.

Additionally, we want to point out that low-level controllers that require consideration of actuator dynamics are not considered in this paper. For coordinated ICC structures, these algorithms are designed and validated by Tier 1 and Tier 2, who design actuators and/or ECUs; together, they ensure the functional safety of the system and perform homologation procedures. In this work, we are focusing on high-level and middle-level controllers.

  1. I posit that chassis control is inherently an engineering task. Nevertheless, the author's review appears to lean excessively towards an academic perspective.

Answer R2.8: We agreed chassis control has been intensively investigated in industry and academia. Since the information about industrial solutions is confidential, we have tried to provide an extensive survey of various approaches from academia. Consiering that the proposed paper is submitted as a journal paper, we believe that it is still valuable for the research community.

Reviewer 3 Report

Comments and Suggestions for Authors

Comments and questions are in attached file. 

Author Response

Dear Editor and Reviewers,

 Thank you for your valuable comments. All of them were taken into account, and the changes are highlighted in the paper. You can find the detailed answers below.

Additionally, text, figures and tables were added to improve the clarity and readability of the paper.

Answers are provided in attached file

Round 2

Reviewer 2 Report

Comments and Suggestions for Authors

The authors have made a commendable effort to address most of my concerns within the available space. The revised version of the manuscript is enhanced noticeably. I am satisfied with the amendments made. I would recommend this paper to be published. 

Comments on the Quality of English Language

Minor editing of English language required

Author Response

Thank you for your positive feedback. The paper was reviewed one more time to improve English.